# Humans parsimoniously represent auditory sequences by pruning and completing the underlying network structure

**Lucas Benjamin\*, Ana Fló, Fosca Al Roumi, Ghislaine Dehaene-Lambertz**

Cognitive Neuroimaging Unit, CNRS ERL 9003, INSERM U992, Université Paris-Saclay, NeuroSpin center, Gif/Yvette, France

**Abstract** Successive auditory inputs are rarely independent, their relationships ranging from local transitions between elements to hierarchical and nested representations. In many situations, humans retrieve these dependencies even from limited datasets. However, this learning at multiple scale levels is poorly understood. Here, we used the formalism proposed by network science to study the representation of local and higher-order structures and their interaction in auditory sequences. We show that human adults exhibited biases in their perception of local transitions between elements, which made them sensitive to high-order network structures such as communities. This behavior is consistent with the creation of a parsimonious simplified model from the evidence they receive, achieved by pruning and completing relationships between network elements. This observation suggests that the brain does not rely on exact memories but on a parsimonious representation of the world. Moreover, this bias can be analytically modeled by a memory/efficiency trade-off. This model correctly accounts for previous findings, including local transition probabilities as well as high-order network structures, unifying sequence learning across scales. We finally propose putative brain implementations of such bias.

**\*For correspondence:**
lucas.benjamin@cea.fr

**Competing interest:** The authors declare that no competing interests exist.

## Editor's evaluation

This paper communicates important findings on the learning of local and higher order structures in auditory sequences and will be of interest to researchers studying statistical learning, learning of graph structures, and auditory learning. The strength of the evidence is convincing, including a compelling demonstration that humans do not encode objective transition probabilities and the implementation of a wide range of sequence learning models that have been proposed in the literature.

## Introduction

"*The fact then that many complex systems have a nearly decomposable, hierarchic structure is a major facilitating factor enabling us to understand, describe, and even 'see' such system and their parts*" – H. Simon, The architecture of complexity (1962).

To interact efficiently with their environment, humans have to learn how to structure its complexity. In fact, far from being random, the sensory inputs we face are highly interdependent and often follow an underlying hidden structure that the brain tries to capture from the incomplete or noisy input it

receives. For instance, *Tenenbaum et al., 2011*, proposed that learning implies building the simpler underlying relational model which can explain the data. Indeed, evidence suggests that humans can infer structures from data at different scales, ranging from local statistics on consecutive items (*Saffran et al., 1996*) to local and global statistical dependencies across sequences of notes (*Basirat et al., 2014*; *Bekinschtein et al., 2009*) or more high-order and abstract relationships such as pattern repetitions (*Barascud et al., 2016*), hierarchical patterns, and nested structures (*Dehaene et al., 2015*), networks (*Garvert et al., 2017*; *Schapiro et al., 2013*), and rules (*Maheu et al., 2020*).

At first, the extraction of local regularities in auditory streams was proposed as a major mechanism to structure the input, available from an early age since *Saffran et al., 1996*, showed that 8-month-old infants can use transition probabilities (TPs) - $P\left(E_t|E_{t-1}\right)$ - between syllables to extract words from a monotonous stream with no other available cues. Since then, the sensitivity of humans to local dependencies has been robustly demonstrated in the auditory and visual domain (*Fiser and Aslin, 2002*) without the focus of attention (*Batterink and Choi, 2021*; *Batterink and Paller, 2019*; *Benjamin et al., 2021*) and even in asleep neonates (*Benjamin et al., 2023*; *Fló et al., 2022*). Moreover, it is not limited to adjacent elements but can be extended to non-adjacent syllables - $P\left(E_t|XE_{t-2}\right)$ - that could account for non-adjacent dependencies in language (*Peña et al., 2002*).

However, the computation of TPs between adjacent - $P\left(E_t|E_{t-1}\right)$ - and non-adjacent elements - $P\left(E_t|XE_{t-2}\right)$ - seems too limited to allow the extraction of higher-order properties without an infinite memory that the human brain does not have. Network science - an emerging interdisciplinary field - thus proposed a different description to characterize more complex streams (*Lynn et al., 2020*). In this framework, a stream of stimuli corresponds to a random walk in the associated probabilistic network. Several studies used this network approach to investigate how humans encode visual sequential information (*Garvert et al., 2017*; *Mark et al., 2020*). *Schapiro et al., 2013*, tested human adults with a network consisting of three communities (i.e. sets of nodes densely connected with each other and poorly connected with the rest of the graph; *Newman, 2003*) where transitions between all elements were equiprobable (each node had the same degree). This community structure is an extreme version of the communities and clustering properties that are often found in real-life networks, whether social, biological, or phonological (*Girvan and Newman, 2002*; *Karuza et al., 2016*; *Siew, 2013*). The authors reported that subjects discriminated transitions between communities from those within communities. Since local properties (TP) were not informative, this result revealed participants' sensitivity to higher-order properties not covered by local probabilistic models. This sensitivity seems already to be in place at 6y-o (*Pudhiyidath et al., 2020*). Recently, *Lynn et al., 2020*, replicated a similar effect with a probabilistic sequential response task. They presented subjects with sequences of visual stimuli that followed a random walk into a network composed of three communities. After each stimulus, subjects were asked to press one or two computer keys, and their reaction time was measured as a proxy of the predictability of the stimulus. To explain the response pattern, the authors proposed an analytical model that optimizes the trade-off between accuracy and computational complexity by minimizing the free energy function. This model allows taking into account the probability of memory errors in the computation of the TPs between the elements of the stream. From now on, we will refer to this model as the free energy minimization model (FEMM: model D, explained below).

In this paper, we aim to merge these two lines of research and validate a model that can explain how humans learn local and high-order relations simultaneously present in sequences generated from noisy or incomplete structures. Moreover, we propose that adults do not encode the exact input but a parsimonious version based on the generalization of the underlying structure. To this end, we leveraged the community network framework and adapted it to expose adult participants to rapid sequences of sounds that followed a random walk through a network, building on the studies described above (*Lynn et al., 2020*; *Schapiro et al., 2013*), but using sparse communities with missing transitions between elements of the same community (see *Figure 1*). This design allows investigating whether participants are able to complete the network according to the high-order structure or whether, on the contrary, they rely on local transitions and reject impossible transitions ignoring the high-order structure. In other words, after training with an incomplete network, if new ('unheard') transitions are presented, are participants more willing to accept them if they belong to the community (i.e. within community transitions) than if they occur between communities? Moreover, while several papers have studied network learning in the visual domain (*Karuza et al., 2019*; *Lynn et al., 2020*; *Schapiro et al., 2013*), to our knowledge, it has never been tested in the auditory domain despite the better statistical

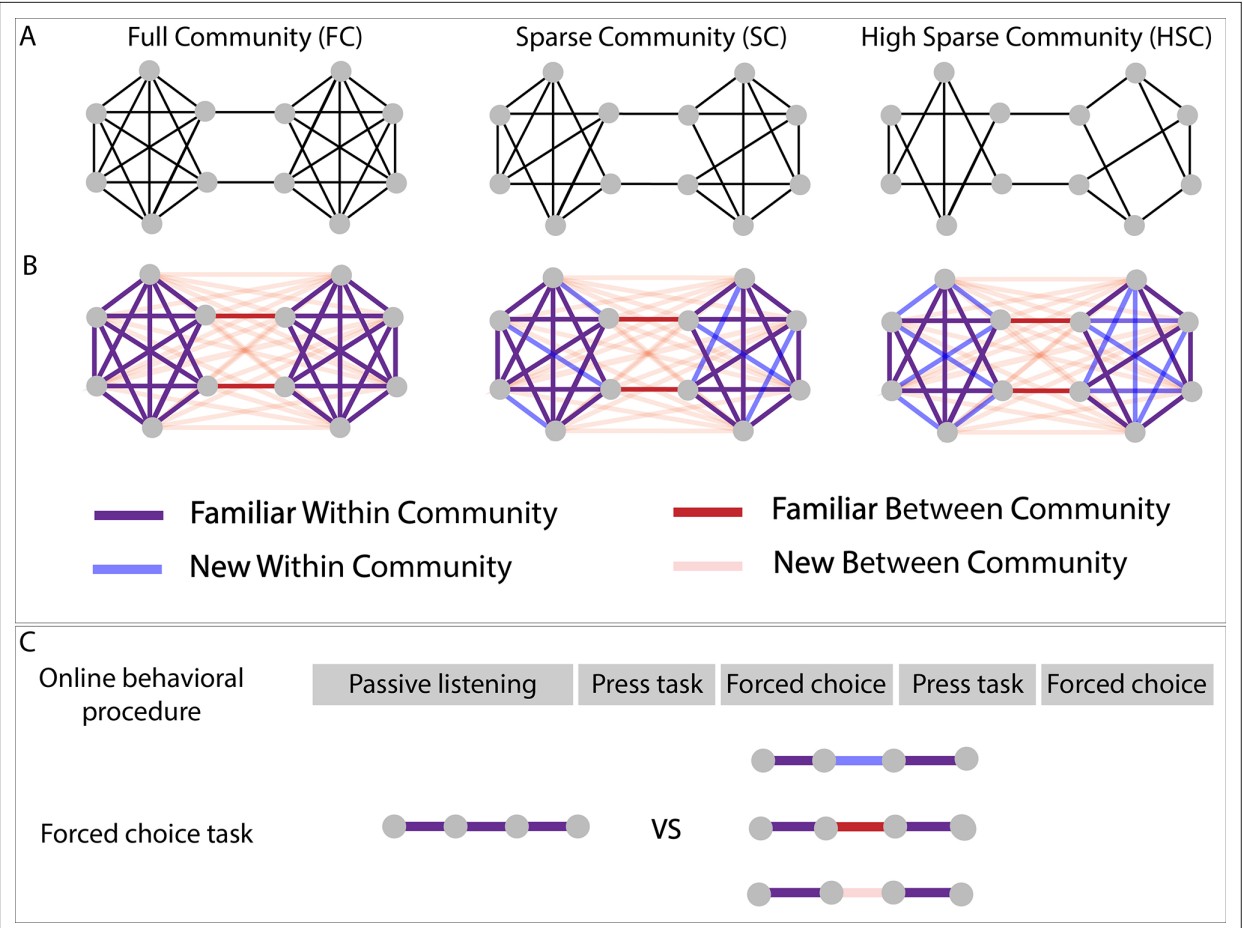

**Figure 1.** Experimental design. (**A**) Graph structure to which adult subjects were exposed in three different paradigms. (**B**) Graph design with color-coded conditions. Blue and pink lines represent transitions that have never been presented during the stream presentation but only during the forced-choice task. (**C**) Test procedure used for behavioral testing. In the press task phase, participants had to press a key when they felt there was a natural break in the sequence. In the forced-choice task, they had to choose between two quadruplets, the most congruent with the sequence they had heard. In the proposed pair, one was always a familiar within condition transition (purple transitions), and the other, one of the three other conditions.

learning capacities in the auditory modality (*Conway and Christiansen, 2005*), the sophisticated auditory sequence processing abilities observed in humans compared to other primates (*Dehaene et al., 2015*), and their potential importance in language acquisition. In addition, the original design was at a very slow rate, allowing for possible conscious decision to take place on the adequation of each element of the sequence to the structure. Here, we used a 4 Hz presentation, typically used in auditory sequence learning tasks, in order to force rapid processing of each element of the sequence and to be more comparable to the sequence learning literature. Finally, we compared how the different models proposed in the literature might fit our data and proposed a unified hypothesis of how any structure (local or global) might be extracted from a sequence.

For this purpose, we tested three different experimental paradigms in an online task, using sequences of pure tones or of syllables (~240 adult participants tested in each paradigm). The first paradigm - full community - tested a network composed of two communities of six elements each, with all nodes within a community connected with each other (except two nodes at the border of the community to keep an equal degree for each node). In the second and third paradigms, the communities were incomplete, some connections being never presented during the exposure to the continuous sequence: In the sparse and high sparse community paradigms, respectively one and two possible edges for each node were removed. The performances in these two 'sparse' designs, relative to the full community design, are crucial to investigate the participants' underlying representations of the sequences.

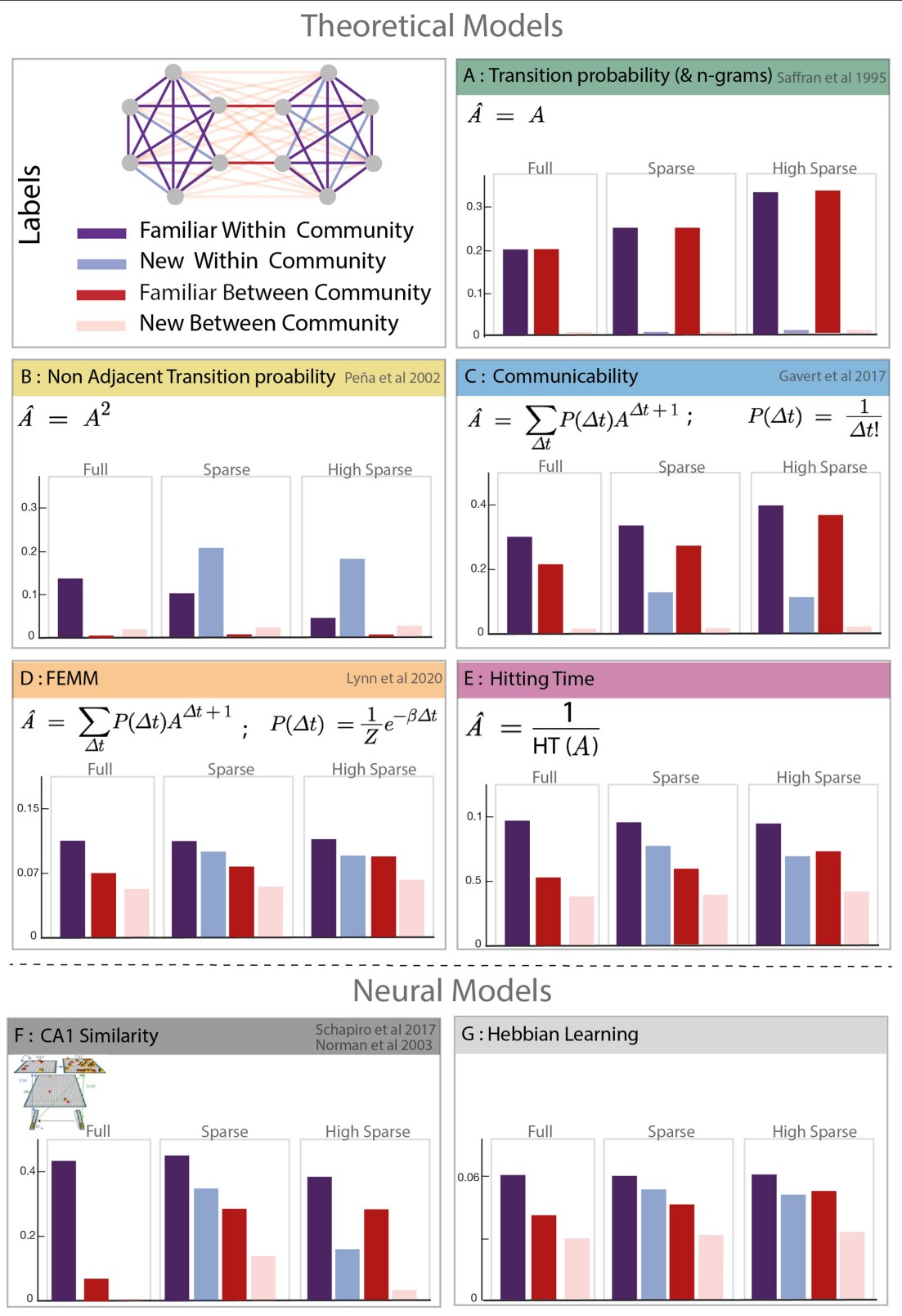

**Figure 2.** Model predictions. Model description and predictions for the three paradigms tested. For each model, we computed the estimated familiarity (a.u.) predicted for each condition in the full, sparse, and high sparse paradigms. Although the models are partially correlated, they differ in their prediction about the familiarity of new within community transitions (light blue) which allows to separate the different models. Models D and E (free energy minimization model [FEMM] and hitting time) are two variations of the same sequence property from a statistical modeling or sequential point

*Figure 2 continued on next page*

*Figure 2 continued*

of view. Their predictions are thus almost identical. Models A, B, C, D, and E are theoretical metrics over the graph structure that predict more or less familiarity with the different types of transitions. Models F and G are biologically plausible neural encoding of those metrics. The box colors correspond to the conditions labeled in the top-left panel.

In each paradigm, participants were first asked to carefully listen to a continuous sequence for about 4 mn and then to press a key when they felt there was a natural break in the sequence (~2 mn). This task allowed measuring participants' ability to parse the sequence and to compare their performances in the auditory domain with those published in the visual domain. In a following test phase, they were asked to choose between two isolated quadruplets, the most congruent with what they had heard before, during the familiarization sequence. With this test phase, we could present previously unheard transitions ('new transitions') and study whether participants were able to generalize the network structure (*Figure 1*), notably in the two incomplete networks (sparse and high sparse paradigms). These two tasks were done twice.

In the forced-choice task between the isolated quadruplets, we tested each other conditions against the *familiar within community transitions* (condition considered as the reference; *Figure 1C*). If participants did not learn the graph structure of the sequence, they had to be random in their familiarity choice between *familiar within* and *between community* transitions because all quadruplets have been presented and had the same local TPs between their elements. By contrast, if they had indeed learned the graph, their familiarity score should be below 50% denoting their preference for the *familiar within community transitions* (i.e. reference). The performances for the unheard transitions, which can be either within or between community transitions (i.e. *new within community* condition and *new between community* condition) relative to the reference should allow to separate the different models proposed in the literature to explain how structures are perceived. Therefore, we compared the participants' behavior (i.e. their familiarity rating for the presented transition relative to the reference) to the predictions of different theoretical models proposed in the stream processing and graph learning literature (*Figure 2*).

- **Model A: TPs and Ngrams:** Local transitions between consecutive elements - $P\left(E_t|E_{t-1}\right)$ - have been proposed as an efficient learning mechanism to structure streams of input. We tested the limits of this simple local learning computation in the presence of a high-order structure. Ngrams are similar to TP but take into account *n* previous items in the computation of the transition. For example, for trigrams, $P\left(E_t|E_{t-1}E_{t-2}E_{t-3}\right)$. Note that because our designs are random walks into Markovian networks, the TPs and Ngram models are identical, $P\left(E_t|E_{t-1}E_{t-2}E_{t-3}\right) = P\left(E_t|E_{t-1}\right)$. Chunking-based models, such as PARSER (*Perruchet and Vinter, 1998*), rely on the repetition of chunks of consecutive elements and, as TP and Ngrams, would reject any chunk with new transitions as they never occurred during familiarization.

- **Model B: Non-adjacent TP:** This metric is similar to the TPs but on non-consecutive items $P\left(E_t|XE_{t-2}\right)$. We included it in our analysis because several studies have shown human sensitivity to such properties in streams (*Peña et al., 2002*).

- **Model C: Graph communicability:** This model comes from the network science literature and computes the relative proximity between nodes in the network, making it sensitive to cluster-like structures like communities. Interestingly, a recent study shows that this measure correlates with fMRI data (*Garvert et al., 2017*), suggesting a potential relevance in human cognition.

- **Model D: FEMM:** This model, recently proposed by *Lynn et al., 2020* to account for community sensitivity by humans, is a trade-off between accuracy and computational complexity. It can be explained by memory errors while computing TP between elements in a stream. Participants exposed to a stream of elements reinforce the association between element i and i-1. However, errors in this process may lead participants to sometimes bind element i with element i-2, i-3, i-4.... with a decreasing probability (for a full description of the model, see *Lynn et al., 2020*). Mathematically, the distribution of the error size that minimizes the free energy function is a decreasing exponential (Boltzmann distribution). Therefore, the estimated mental model of TP is biased compared to the streams' objective TPs enabling participants to encode high-order structure. In more detail, the mental model is a linear combination of the TP matrix (A) and non-adjacent TPs of every order ($A^{\Delta t}$) with a weight of P ($\Delta t$) where $\Delta t$ is the order of non-adjacency

(or size of the memory error, i.e. Δt = n corresponds to P (E$_t$|X.....XE$_{t-n}$)). The estimated model can then be written as:

$$\hat{A} = \sum_{\Delta t=0}^{+\infty} P(\Delta t) A^{\Delta t+1}$$

with

$$P\left(\Delta t\right) = \frac{1}{Z} e^{-\beta \Delta t}$$

where $A$ is the TP matrix of the graph. $\beta$ was previously estimated to 0.06 in a comparable task with human adults (**Lynn et al., 2020**). We therefore first used this value to test this model on our behavioral data and later confirmed this estimation with our data (see SI). In the reinforcement learning literature, the hippocampal place cells have been proposed to represent maps of probabilistic future states and reward by encoding *successor representation* instead of positional cognitive maps (**Dayan, 1993**; **Stachenfeld et al., 2017**). Successor representation has been formally defined as the sum of probabilistic future state and can be written SR $= \sum_{\Delta t} \gamma^{\Delta t} A^{\Delta t}$.

This approach is very similar to FEMM with an infinite sum of all power of the transition matrix, pondered by an exponentially decreasing factor. Here, the factor is $\gamma^{\Delta t}$ with $0 < \gamma < 1$ and generally $\gamma = \frac{0.85}{\lambda max}$ with $\lambda max$ the largest eigenvalue of the transition matrix (**Garvert et al., 2017**). This approach has been proposed to account for community perception (**Pudhiyidath et al., 2022**) but here we only included FEMM in our study, as the two models are identical with $\gamma = e^{-\beta}$ (with a different constant).

Another metric computing the same property but from a sequence point of view is the hitting time.

- **Model E: Hitting time:** This metric, also coming from network science, estimates the distance between two nodes in a graph as the average number of edges needed (path length) to move from one node to another during a random walk. Similar to communicability (model C) and FEMM (model D), it measures a 'proximity' between nodes in a network. To make it more comparable with the other models, we computed its inverse value.

Although the different models are partially correlated with each other, they give different predictions about participants' familiarity responses. First, they were two kinds of local transitions: familiar transitions and new transitions (TP = 0). Since the TP calculation does not consider the community structure (model A), participants should equally reject new transitions regardless of their relation with respect to communities (new within communities = new between communities). Second, concerning the new transitions, FEMM and hitting time models predict that participants should better detect *new between community* than *new within community transitions (completion effect)*. It is also partly the case for the communicability model, but not for the TP and non-adjacent TP models (models A and B). The similarity of the predictions of FEMM, hitting time, and communicability models is not surprising as they all describe the same property of the network: proximity between nodes. Intuitively, items from the same community will appear closer together than items from different communities, even if the two nodes are not connected. In fact, FEMM and communicability are mathematically very close but with a different decay (exponential vs. factorial). However, they can still be differentiated thanks to the high sparse paradigm were the relative predicted familiarity of *new within* and *familiar between* transitions are different between the two models.

In addition to those theoretical models, we considered two putative brain implementations using biologically realistic neural networks:

- **Model F: Hippocampus CA1 similarity:** This neural network aims to reproduce the hippocampus structure (**Norman and O'Reilly, 2003**), which is often described as a key structure in statistical and structure learning (**Henin et al., 2021**; **Schapiro et al., 2017**; **Schapiro et al., 2016**). We compute here the similarity in CA1 layer as it has been proposed to capture community-like structures in previous studies (**Schapiro et al., 2017**). Indeed, thanks to its ability to have overlapping representations of the input and direct connection with the entorhinal cortex through

the monosynaptic pathway, CA1 structure is also sensitive to long-distance dependencies allowing high-order structure learning.

- *Model G: Hebbian learning with decay:* Hebbian learning is a biologically plausible implementation of associative learning. Some neurons fire specifically to some objects in the environment. When two of those neurons co-fire, the pair is reinforced. It has been suggested that learning TPs is based on such a mechanism in the cortex. Here, we adapted this idea to implement the FEMM computation instead of TP, specifically by adding a temporal exponential decay in the probability of a neuron firing after a stimulus's presentation. When the exponential decay has the same β parameter as the FEMM, the results of the FEMM and the Hebbian learning with decay are mostly similar.

## Results

### Human behavior

#### Key presses distribution during active listening

All participants were exposed to a stream of either tones or syllables adhering to one of three possible graphs (*Figure 1A and B*). After a 4 mn familiarization period, they were instructed to press the spacebar when they felt the impression of a natural break in the sequence (2 mn). This task was a sanity check to corroborate that participants were listening to the stream and that their performance was comparable to previous studies testing graph learning using the visual modality at a much slower pace than we used here. *Figure 3* top row shows the normalized distribution probability of key

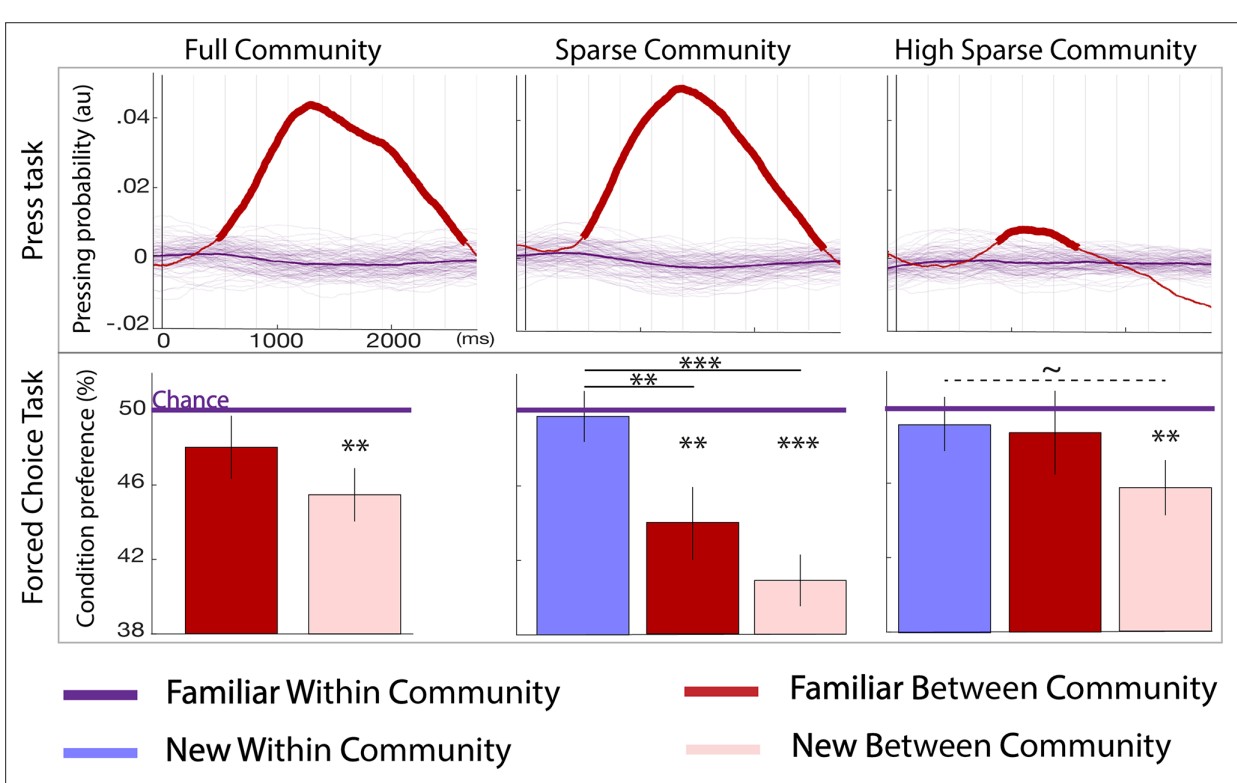

**Figure 3.** Behavioral results. Top panel: parsing probability during the active listening phase (distribution of key presses after the offset of a given transition) purple lines: familiar within community transitions, red line: familiar between community transitions. Thin purple lines each represent a bootstrap occurrence of the parsing probability for the familiar within community transition. The bold red line indicates the time points where there was a significant increase of parsing probability after a familiar between community transition compared to a familiar within community transition. Bottom panel: familiarity measure in each paradigm: percentage of responses for each condition during the forced-choice task. By design, the chance level (50%) represents the familiar within community estimated familiarity (reference). The stars indicate significance against the reference and between conditions (pval <0.05 FDRcorr) the dotted line marginal significance (pval = 0.046 uncorr). The error bars represent the standard error for each condition. N=728 participants were tested to the Full Community (N = 250), the Sparse Community (N = 249), or the High Sparse Community (N = 228) paradigms.

presses after a transition, using a kernel approach (see Materials and methods for detailed computation). In all three paradigms (each corresponding to a graph in *Figure 1*), the significant increase in key presses after between community vs. within community transitions (p<0.05 are indicated in bold lines) reveals that participants were sensitive to the switch between sound communities. Full community and sparse community designs showed a similar effect size, while the high sparse community design elicited a small but significant effect. Unpaired t-tests every ms in [–0.1, 2.750] s window, contrasting the full community vs. high sparse community, show a significant difference between 1 and 2.6 s post-transition (p<0.05 Bonferroni corrected). Similarly, sparse community vs. high sparse community differed between 0.8 and 2.5 s (p<0.05 Bonferroni corrected).

### Two-forced-choice task

Participants were given a two-forced-choice task, in which they had to choose between two sequences the one that best matched the structure of the stream they had listened to (*Figure 1C*). This task is the crucial test for comparing models because it allows to present new transitions that matched, or not, the familiar structure and thus to assess the representation of the memorized graph. We report the results at the end of the learning (second block). Results separated by groups and testing block are presented in SI. It can be seen that in contrast with the three other data points, participants' choice were close to random after the first block in the syllable experiment and their performance could not be explained by any of the models. As pointed in other experiments on statistical learning using syllables (*Elazar et al., 2022*; *Onnis and Thiessen, 2013*; *Siegelman et al., 2018*), the familiarity with speech and the phonetic rules of the native language create priors on the probability of sequences of syllables, that might compete with the real syllable distribution in the task. At the end of learning, no difference was found between the groups using tones and syllables (unpaired t-test for each condition, all ps >0.2), we thus merged the data of the tone and syllable groups.

In this task, scores below 50% indicate that the reference (*familiar within community* transitions) was judged more familiar than the tested condition. We postulated that if participants were only sensitive to familiar transitions, any novel transitions should be judged less familiar than the *familiar between community* transition. On the other hand, if participants encoded the underlying structure of the communities, they should not notice the novelty of the *new within community* transitions and reject the two between community conditions (familiar and new).

As can be seen in *Figure 3*, participants significantly rejected the *new between community* transitions in each paradigm (ps <0.01 FDR), this transition is both novel and jumping across communities. The *familiar between community* transition condition was only significantly rejected in the sparse community paradigm (p<0.01 FDR). Second, the *new within community* transitions were chosen/ rejected at chance in the sparse and high sparse community paradigms indicating a similar perception of familiarity for these never heard transitions and the reference. Third, in the sparse community paradigm, the familiarity score was larger for the *new within community* transitions than for both between community transitions (new: p<0.01 FDR; and familiar: p<0.05 FDR). These comparisons were only marginally significant in the high sparse paradigm (uncorrected p=0.046). In other words, the participants encoded the graph structure as revealed by the difference in familiarity between within and between community transitions and naturally completed the graph as indicated by the scores at chance for never heard transitions compatible with the graph structure.

## Which model best fits the participants' behavior

### Correlation between human data and theoretical model predictions

To estimate the adequacy of the theoretical models to explain the behavioral data, we pooled together the three paradigms and estimated the correlation with each model. We normalized each model prediction by the model's value for *familiar within community* transitions to be comparable with the behavioral results of the two-force-choice task. It is worth noticing that models A, B, C, and F predict differences in familiarity for familiar within transitions between the three paradigms (full, sparse, high sparse); however, our experimental design does not allow us to estimate differences in these transitions between paradigms but only relative differences to the *familiar within community* condition within paradigms. To estimate the significance of the correlation differences, we used a bootstrapping approach with subjects (with replacement) and estimated the number of bootstrap occurrences in favor of one model against another. *Figure 4A* shows the correlations' distribution

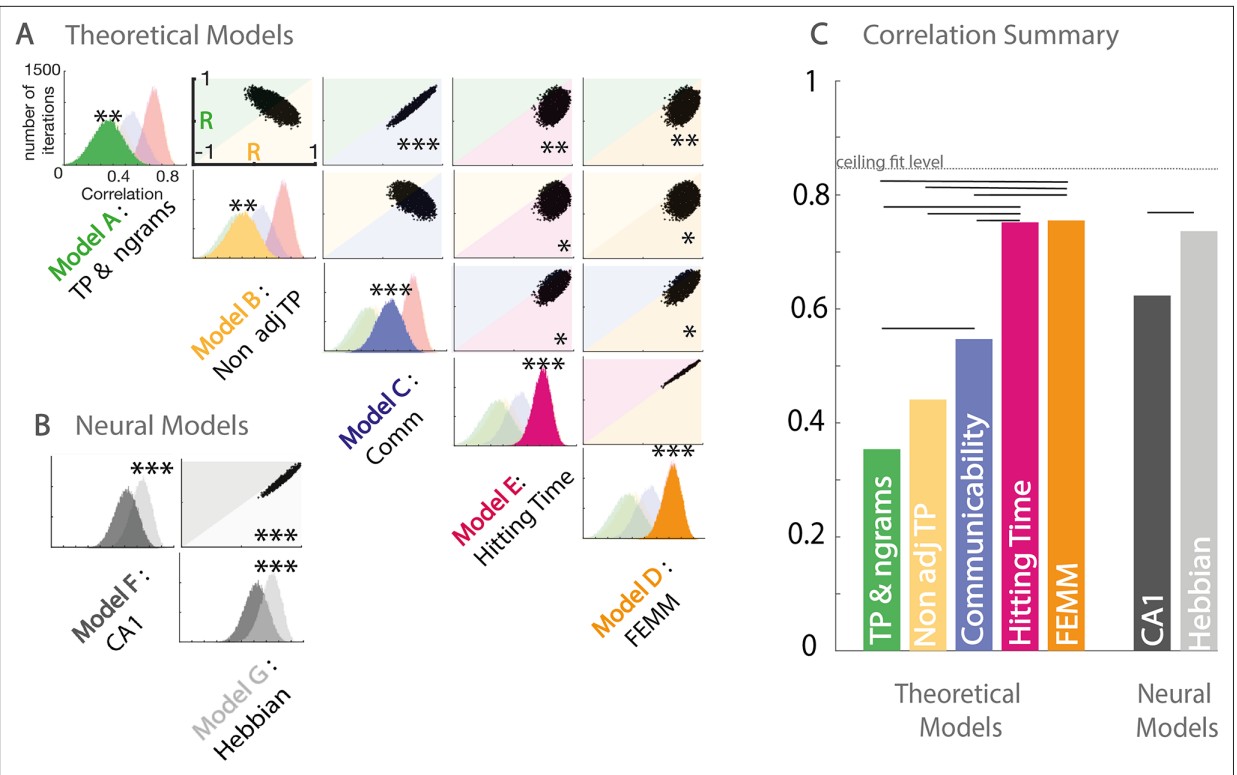

**Figure 4.** Model and data comparisons. (**A**) Estimation of the correlation of the participants' familiarity score pattern with each theoretical model (**A to E**) using bootstrap re-sampling. The diagonal of the matrix displays the distribution of correlations between the participants' familiarity pattern across conditions and the predictions generated by each model (**A**), theoretical models (**A to E**), and (**B**) neural models (**F&G**). Each panel of the diagonal presents the same result, the color of the relevant model being highlighted to facilitate the comparison between models. For each pair, the significance between models (indicated by stars) is estimated by counting the number of bootstrap occurrences for which one model was more correlated with the data than the other. We plotted this bootstrap as a cloud of dots in the Correlation with Model1 × Correlation with Model2 subspace. Significance is then represented by the percentage of dots above the diagonal. Models with similar predictions display a line style cloud of dots aligned along the diagonal. (**B**) We did the same comparison with the two neural models (**F&G**). (**C**) Summary of the correlations between each model and the behavioral data. Plain lines above the boxes indicate the significant differences between models. FEMM and hitting time (**D&E**) are equivalent and equally good and significantly better than all other theoretical models. For neural models, the Hebbian model (**G**) shows a slight, but highly significant, better fit with the participants' scores. The dotted line indicates the ceiling fit level estimated for this dataset.

The online version of this article includes the following figure supplement(s) for figure 4:

**Figure supplement 1.** Group by group analysis.

**Figure supplement 2.** Correlation analysis on contradictory predictions.

between the data and each model (presented on the diagonal) and between pairs of models. We estimated the significance of the correlation strength between the data and model *i* or *j* by counting the percentage of occurrences in which model *i* had a stronger correlation with the data than model *j*. All models were significantly correlated with the data (all p<0.01 FDRcorr), with a correlation strength following the order FEMM ≈ hitting time > communicability > non-adjacent TPs ≈ TPs (*Figure 4C*). Note that the FEMM and hitting time are similar models, and thus predictions are almost identical. They had the best correlation with the data (81%) and were significantly better than all the other theoretical models (p<0.05 FDR).

## Correlation between human data and neural model predictions

As the FEMM computation and the hitting time were the best theoretical models, we translated them into a realistic biological architecture using Hebbian rules. We estimated this implementation on a 50,000 item-long stream for each paradigm. The correlation between the analytical computation and the Hebbian learning implementation exceeds 99%. Using the same bootstrap approach, we compared this Hebbian approach with a neural network reproducing hippocampus architecture

proposed by *Norman and O'Reilly, 2003*. Both models were significantly highly correlated with the data and with each other. However, the Hebbian implementation of FEMM was slightly but significantly more correlated to our data than the hippocampus model (*Figure 4*) typically because of the lack of agreement between the hippocampus model and the data in the high sparse paradigm. However, because the hippocampus model highly fits our data, we cannot rule out the hippocampus as a potential crucial structure for such tasks.

## Estimation of the ceiling correlation with our data

We also used the same bootstrapping approach to estimate the noise ceiling for the model fit. For each bootstrap, we randomly selected *n* subjects with replacement twice and correlated the data of those two random samples. We find an average of 84% correlations as a noise ceiling for those data. Our best fit with any model is the 77% average bootstrap correlation between our data and the FEMM, which is relatively close to the ceiling fit given this dataset, showing a very high relevance of the FEMM to account for the data.

# Discussion

## TPs between elements of the sequence are biased by the structure of the underlying generative network

Our results show that human adults do not encode TPs objectively when familiarized with a stream of sounds. Instead, they seem to have a systematic bias to complete the transitions within a community suggesting a subjective internal representation that differs from the objective distribution of the transitions they heard. This behavior is compatible with two proposed theoretical models: the FEMM and the hitting time.

The high agreement between the FEMM and the data we observed suggests that the bias can be analytically estimated using the FEMM $\hat{A} = \sum_{\Delta t=0}^{+\infty} P(\Delta t) A^{\Delta t+1}$ with $P(\Delta t) = \frac{1}{Z} e^{-\beta \Delta t}$. *Lynn et al., 2020*, proposed that this bias corresponds to memory errors when recalling the previous item of the stream during the TP computation. The bias in the encoding of TP between successive elements enabled the extraction and encoding of high-order structures in graphs, that is, a community structure. We can distinguish two distinct bias effects: First, the *pruning* of familiar transitions that do not conform to the community structure (i.e. *familiar between community* transitions are rejected). Second, the *completion* of the structure by overgeneralizing new transitions when they are compatible with the high-order structure (i.e. *new within community* transitions are accepted). These perceptual biases lead to a more parsimonious internal representation of graphs.

## Putative brain implementation of such computation

We showed that the computation of TPs is biased in humans, and analytically, this bias is characterized as an optimal trade-off between accuracy and computational complexity. Indeed, perfect accuracy in the encoding would result in no sensitivity to the high-order structure, while too low accuracy would result in no learning at all. We also presented putative brain implementations and tested to what extent two previously described mechanisms might explain our results: Hebbian learning and hippocampus episodic memory.

Hebbian learning is a very simple mechanism that consists of reinforcing co-occurrences in a signal. It has been proposed as a learning mechanism in statistical learning tasks (*Endress and Johnson, 2021*). Here, we minimally modified it as described above to introduce the bias in TP computation. Such learning could be implemented in many brain regions through learning-induced synaptic plasticity and does not require any specific structural organization of neurons. In contrast, the CA1 similarity model relies on the specific architecture of the hippocampus. Testing a hippocampus specific model is essential because several authors have proposed that statistical learning and graph learning might be represented as the construction of an abstract map of relational knowledge, analogous to topographic maps (*Constantinescu et al., 2016*; *Garvert et al., 2017*), which are known to involve the hippocampus. Moreover, the hippocampus has also been proposed as a good candidate for the implementation of the successor representation, giving this structure the role of a predictive map unifying temporal and spatial relational knowledge under a common framework (*Stachenfeld et al., 2017*).

A recent experimental study (*Henin et al., 2021*) showed that when exposed to statistically organized auditory or visual streams, the hippocampus activity measured with ECoG exhibited a cluster-like behavior, with all elements belonging to the same group being similarly encoded. Using the community paradigm with fMRI, *Schapiro et al., 2016*, also reported an increased pattern similarity in the hippocampus for elements belonging to the same community (see also *Pudhiyidath et al., 2022*). Another piece of evidence comes from modeling the hippocampus activity in different statistical learning tasks (*Schapiro et al., 2017*). In this study, the authors used a neural model mimicking the hippocampus architecture and trained it on different statistical learning tasks including community structure learning. They showed that the pattern of activity in CA1 might account for both pair learning (episodic memory) and community structure learning, and thus is partially consistent with two mechanisms observed in the hippocampus: pattern completion (i.e. the similarity of the neural representations of close stimuli increases, which allows generalization) and pattern separation (i.e. the similarity of neural representation of close stimuli decreases, to disambiguate them) (*Bakker et al., 2008*; *Liu et al., 2016*; *Yassa and Stark, 2011*).

Here, we showed that both a general Hebbian model and a more specific hippocampal model fit very well the pattern of familiarity scores given by the participants with a slightly better result, yet significant, for the Hebbian learning approach. Since we only have behavioral results, it is difficult to conclude on the exact brain regions involved, especially since recent work proposed the joint use of several computation involving cortical and hippocampal learning in similar tasks (*Varga et al., 2022*; *Whittington et al., 2020*). In any case, the agreement between the behavioral data and two brain models shows that the FEMM (an analytical model) does not only explain behavioral data but also has biologically valid candidates.

## A general model of statistical learning for sequence acquisition

Statistical learning has been proposed as a powerful general learning mechanism that might be particularly useful in language acquisition in order to extract words from the speech stream (*Saffran et al., 1996*). However, the exact model explaining statistical learning remains under-specified: What is computed remains unclear (*Fló et al., 2022*; *Henin et al., 2021*) and authors often tailored the computation to suit the paradigm (TPs in some studies, non-adjacent or backward TPs in others, biased transitions probabilities in network studies, etc.). We argue that the FEMM is a more general model that, beyond explaining community separation, as shown above, can also account for results traditionally explained by the computation of local transitional probabilities and those that require the computation of long-distance dependencies. Indeed, the first-order approximation of the FEMM corresponds to the objective TPs model ($\hat{A}_0$, see SI). Thus, the predictions of the FEMM are the same as those of the TP model in many tasks, notably in classical speech segmentation experiments, where a drop in TP signals word edges (*Saffran et al., 1996*). Another approach in the literature about sequence learning considers the recognition of chunks more than statistical learning as a primary mechanism for segmenting sequences. Based on this approach, PARSER and TRACXS detect often occurring chunks in sequences but do not associate a familiarity rating with each transition. In a previous experiment (*Benjamin et al., 2023*), we showed that familiarity based on statistical learning does not always lead to sequence chunking and here we focused on this sense of familiarity which does not require the construction of a repertoire of possible chunks postulated by chunking models. Therefore, we did not consider these models here.

Another part of the statistical learning literature focuses on AxC structures, in which the first syllable of a triplet predicts the last syllable (*Buiatti et al., 2009*; *Endress and Johnson, 2021*; *Kabdebon et al., 2015*; *Marchetto and Bonatti, 2015*; *Peña et al., 2002*). The computation of first-order TPs is insufficient to solve this task, which requires the encoding of non-adjacent TPs. However, a bias estimation of TPs following the FEMM is sensitive to non-adjacent dependencies and can explain the emergence of AxC structures. Additionally, as previous papers and our results show, the FEMM can also explain subjects' behavior in different kinds of network learning (*Karuza et al., 2016*; *Lynn et al., 2020*; *Schapiro et al., 2013*). *Lynn et al., 2020*, interpret the FEMM as errors in the associations between elements, whose probability decays with the distance between associated elements. We proposed that implementing the TPs computation through Hebbian learning with a firing decay results in a comparable computation to the free energy model.

Finally, a similar Hebbian learning approach enables to explain the sensitivity to backward TP reported in the literature (*Endress and Johnson, 2021*; *Pelucchi et al., 2009*). A similar idea has recently been proposed by *Endress and Johnson, 2021*. However, the authors did not refer to free energy optimum or provide an analytical approach. Instead, they proposed a Hebbian learning rule with the same idea of mixing TP with non-adjacent TP (which corresponds to a second-order approximation of the FEMM that we propose here, see $\hat{A}_1$ in SI). Like we do here, they argued that this mechanism could account for results currently explained by different models in the literature. Thus, the FEMM and its putative neural implementation through Hebbian rules unifies different proposals concerning statistical learning on the one hand and network learning results on the other hand, under a common principle. It is important to note that we investigated how the FEMM - and the other models - accounts for the extraction of regularity from a sequence, which is the first needed step of many other processes. We did not test for further abstract representations of the sequence that could be subsequently computed.

## Information compression and stream complexity

Our results showed that adult humans have a biased subjective representation of first-order TPs compared to the actual TPs, which makes them to be sensitive to high-order structure in the underlying graph and to overgeneralize transitions that they never experienced. What is the advantage of such a computational bias for human cognition? We postulate three main advantages.

Higher-order structures and generalization can be relevant information to learn. Unlike random networks, many real-world networks have transitivity properties (*Girvan and Newman, 2002*; *Newman, 2006*; *Newman, 2003*) - if A is connected to B and B to C, there is a high chance for A and C to be connected (a friend of my friend is likely to be my friend).

Overgeneralizing enables faster learning. Overgeneralizing means accepting transitions congruent with the structure even before they appear in the stream. Thus, for short exposures, the estimation of the FEMM is closer to the real TP matrix than the estimation of the TP model based on the input because it infers transitions that have not been presented yet. This fast learning might be of importance, for example, for language acquisition, given that human infants are exposed to a limited amount of speech.

Adding to why humans have biased statistical learning, we propose that this learning bias in extracting statistical information might subsequently be used to form abstract condensed network

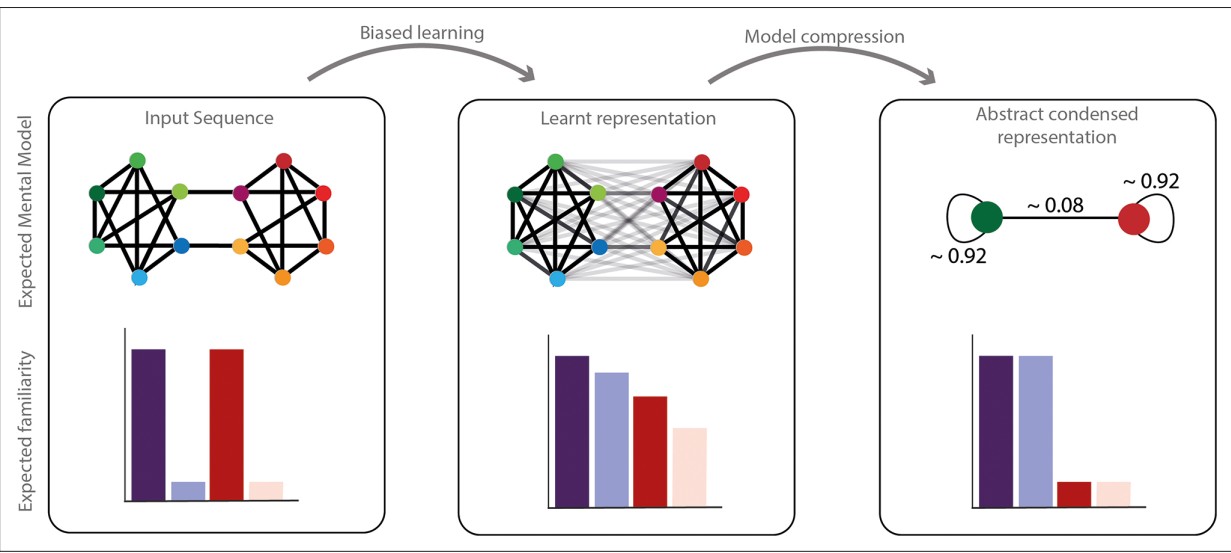

**Figure 5.** Network compression hypothesis. Compressibility hypothesis. In the left panel, the real underlying structure of the input presented. In the middle the learned representation by humans. As described above, this representation does not completely reflect the real input structure but a biased parsimonious version of it, including pruning and generalization of transitions. In the right panel, we hypothesized a condensed representation that might be formed subsequently to simplify and compress the information. In this representation, the identity of the elements would be ignored in favor of their community label. The familiarity of each transition is represented with transparency of the edge in the network representation and each condition familiarity pattern is represented with barplots below.

representations. In fact, the extraction of high-order structures might enable information compression in long-term memory. Because of the computational cost and the pressure on memory to encode long sequences, compressing information is a major advantage. In a community paradigm, the learned representation could be later simplified to reduce the stream complexity to a binary sequence with a certain probability of changing between communities A and B (*Figure 5*). Instead of remembering all the transitions of the stream, remembering community labels and the probability of transition between communities is sufficient. Recent data (*Al Roumi et al., 2021*; *Dehaene et al., 2014*; *Planton et al., 2021*; *Sablé-Meyer et al., 2022*; *Sablé-Meyer et al., 2021*) showed that in some circumstances, humans' performances were highly sensitive to input compressibility, arguing for a condensed encoding of inputs. Note that the familiarity measure we report here does not show compression of the structure. Still, the familiarity bias could be at the basis of a later abstract condensed network representation (this hypothesis is presented in *Figure 5*). In the same line, a recent study using a graph perspective (*Whittington et al., 2020*) proposes that the representation of the abstract relational structure of a sequence and the mapping between node and stimuli identity could be factorized. In the case of community paradigm, *Pudhiyidath et al., 2022*, even proposed that the formation of such an abstract structure could allow humans to transfer learnt properties between elements belonging to the same community. *Mark et al., 2020*, showed that the learning of the structure of a network could be re-used on the next day to allow fast and generalizable learning arguing for a factorized brain representation between the stimuli mapping and the abstract network encoding. This compressibility hypothesis, represented in *Figure 5*, needs formal testing to be confirmed or infirmed.

Finally, the human sensitivity to community is in line with Simon's postulate that the complexity of a system can only be handled, thanks to its hierarchical nearly decomposable property (*Simon, 1962*). In other words, a complex structure is no more than the sparse assembly of less complex dense substructures. Here, we propose empirical arguments by demonstrating that human adults are sensitive to the decomposition of a complex network into two simpler sub-networks.

## Methodological remarks

In this study, we used two different metrics. The press bar task during attentive listening showed high sensitivity, but it only allowed testing within vs between community transitions during learning and thus assessing clustering (different perception of *familiar within* and *familiar between* transitions). The forced-choice task on the isolated quadruplets allowed testing for more conditions after learning and thus to distinguish between models. However, this second metric had a low sensitivity because only a few trials could be collected resulting in high error variance that was compensated by a very large sample of participants (*N*=727).

This design also did not allow us to efficiently study the dynamics of learning. We had only two points for the estimation of the learning of the graph by explicitly detecting quadruplets familiarity. This is particularly insufficient when, as here, the speech or non-speech nature of the stimuli modulate performance because of different priors on the possible composition of the sequences. Even for tones, we could not determine when learning took place as it seems stable from the first measure point.

## Materials and methods
### Behavioral task
#### Participants

A total of 727 French adults were recruited via social media (424 of which were retributed $2.5 on Prolific platform). They had to have no hearing or language problems and French had to be their first language. They were assigned to one version of the experiments and instructed to carefully listen for 4.4 min to a nonsense language composed of nonsense words that they had to learn because they would have to answer questions on the words afterward. Participants were either exposed to the full community (*N*=250), the sparse community (*N*=249), or the high sparse community (*N*=228) paradigms with either pure tones or syllables as stimuli.

## Ethic approval

All participants gave their informed consents for participation and publication and this research was approved by the Ethical research committee of Paris-Saclay University under the reference CER-Paris-Saclay-2019-063.

## Stimuli

We generated 12 tones of 275 ms duration, linearly distributed from 300 to 1800 Hz. We also generated syllables with the same duration and flat intonation using the MBROLA text-to-speech software (*Dutoit et al., 1996*) with French diphones. There was no coarticulation between syllables.

Each experiment was composed of 4.4 min of an artificial monotonous stream of concatenated tones (or syllables) without any pause, resulting from a random walk into the tested graph. The graph was either complete (full community), with one missing transition (sparse community), or two missing transitions at each node (high sparse community) creating three experimental paradigms. To avoid any putative acoustical bias, we collected eight groups of subjects for each paradigm. For each of the eight groups, we randomly generated a new graph (except for the full community graph, for which only one graph was possible), a new correspondence between the alphabet of tones (or syllables) and the nodes of the graph and finally new random walks into the graph.

In the original study (*Schapiro et al., 2013*), the authors explored different graph traversal: random walk and Hamiltonian path. In the Hamiltonian path, each node is presented only once, avoiding short distance repetitions and thus controlling for a putative novelty effect when there is a change of community which could potentially serve as a parsing cue in a random walk. However, participants did not parse the sequences better in the case of random walks relative to Hamiltonian walks (Figure 2 in *Schapiro et al., 2016*) minimizing the concern of a possible habituation effect if random walks are used. Here, we chose a random walk because the Hamiltonian path introduces more predictability to the sequence. As previously presented stimuli of the community can no longer be presented, the predictability of the next element increases with the length of the path within a community until a perfect predictability for the fifth and sixth elements (node at the border of communities) and the next element in the other community whereas a random walk keeps the prediction flat. Thus, learning a graph through a Hamiltonian walk can be fully explained with Ngram approaches and cannot disentangle the different learning models proposed. Moreover, the number of Hamiltonian paths available drastically decreases with sparsity up to the point where, in the high sparse paradigm, a single sequence is possible of a given first element leading to a trivial pattern of repetition of 12 elements.

With a random walk, the tones belonging to the same community are presented on average closer in time than those belonging to different communities. However, the length of the walk within one community can be short without repetition or without going through all the tones of the community, or longer with repetition of some tones at a random distance. Therefore, there is no consistency over time that could allow to capture a repetition pattern. Furthermore, the absolute frequency of each tone is equal within the stream, which avoids long-term habituation effects, and the local TP is flat, which avoids the possibility of predicting the next tone. Finally, the tones frequency was distributed between the two communities, to prevent a separation based on an auditory spectral partition. However, due to the design reasons explained before, Halmitonian walks are not usable and thus we could not formally control for potential habituation effect in our design. The key-press results of this study (but not the two-forced-choice results) are therefore potentially subject to confounding by habituation.

For the isolated quadruplets, we concatenated four sounds so that the first and last transition were always non-deviant (familiar within transition) but that the transition in the middle would be of each type of transition. We used quadruplets in this study for consistency with previous work of the team and especially for comparing latencies of developmental ERPs in possible future electrophysiological work.

## Procedure

Participants started with a 4.4 mn familiarization phase of exposure to the stream (960 items). Then learning was tested with two tasks. First, participants were told that the order of the tones/syllables was not random and that they had to press the spacebar when there was a noticeable change in the tones (or syllables) group used in the stream. Second, they were presented with a two-forced-choice

task in which they had to choose between two quadri-elements sequences, the most likely sequence, part of the language they learned.

The two-forced-choice trials always comprised a *familiar within community* transition and one representing the other conditions. These conditions were *new within community* transitions, *new between community* transitions, and *familiar between community* transitions (*Figure 1*). Participants were exposed to eight trials per type (with different sounds each time) except for the *new within community* type, where they were only exposed to four trials because, by design, there are only four of those transitions in the graphs. Each transition used in the set was presented in both directions (AB and BA). Four catch trials were also included to control participants' engagement in the task. These catch trials were two consecutive identical quadruplets that subjects had to detect. Then, they were again exposed to a random walk stream for 2.2 min (active listening - 479 transitions) followed by the same forced-choice task as before.

## Data processing: active listening task

Participants who pressed less than 10, or more than 200, times during the experiments were excluded from further analysis (FC: 52/250; SC: 24/249; HSC: 23/228). A null array of the stream size was built and filled with ones at times when participants pressed the spacebar (Dirac impulses). To convert it into a continuous signal, we convoluted it with an exponential window. Then, we epoched this continuous signal from –2.75 to 2.75 s after each transition's offset. Finally, we averaged all the epochs corresponding to the four familiar between community transitions and four out of all familiar within community transitions, and compared them. We repeated this with 1000 random groups of four familiar within community transitions in each subject. By normalizing and averaging across subjects, we were able to estimate the increase of the pressing probability after a familiar between community transition compared to a familiar within community transition at each time point. This method is similar to the kernel approach for estimating probability density from discrete observations.

## Data processing: forced-choice task

Participants that failed on more than two catch trials (two identical quadruplets) out of four were excluded from further analysis (FC: 35/250; SC: 45/249; HSC: 34/228). For each subject, we computed a percentage of preference for the tested transition relative to the reference (*familiar within community* transition) in each condition (i.e. the ratio between the number of trials where the subject chose the tested sequence and the total number of trials of this condition). The measure ranges from 0 (the *familiar within community* transition is always selected) to 100 (the other transition is always selected) with a chance level of 50%. We estimated the familiarity score of each condition vs the chance level (50%) using paired t-tests. We report the data from the second forced-choice-task session, corresponding to the maximum exposure to the streams. For the tone stream, results were similar in the first and second sessions. For the syllable stream, results from the first session were poorly consistent across participants, probably because the task was more difficult in the case of syllables. Indeed, flat transitions between syllables violate language structure and participants' priors on syllable sequences. The conflict between priors and the real structure of the sequence might need a variable time to be resolved by each participant (*Elazar et al., 2022*; *Lew-Williams and Saffran, 2012*; *Onnis and Thiessen, 2013*; *Siegelman et al., 2018*). For completeness, we performed the correlation analysis with each subgroup of data (first vs. second session and tones vs. syllables). These analyses are presented in *Figure 4—figure supplement 1*. None of the models could adequately explain the first session of the syllable group. To further investigate the learning dynamics and in particular the influence of priors, another paradigm should be proposed, which is beyond the scope of the present study.

## Modeling
### Theoretical models
For the four models that could be analytically computed from the TP matrix (A, B, C, and E), we computed the predictions made by the models for each of our graphs (eight with syllables, eight with tones). Given $A$ the transition matrix of the graph, models were computed using the analytical description:

- ***Model A: TP and Ngrams:***

  By construction of the transition matrix, the TPs between nodes are the elements of *A*.
  $$\hat{A} = A$$

- ***Model B: Non-adjacent TP:***

  Non-adjacent TPs are computed by taking the square of the transition matrix
  $$\hat{A} = A^2$$

- ***Model C: Communicability:***

  $$\hat{A} = \sum_{\Delta t=0}^{+\infty} P\left(\Delta t\right) A^{\Delta t} \quad with \quad P\left(\Delta t\right) = \frac{1}{\Delta t!}$$

  Thus, $\hat{A}$ corresponds to the exponential series: $\hat{A} = e^A$. We use Matlab function 'expm' to compute this value.

  The communicability model as described in *Garvert et al., 2017*, uses the adjacency matrix. Here, we used the TP matrix. We believe it is more appropriate to consider the relative weights of each transition and not only its existence or not, because a random walk into a weighted graph follows the transition matrix and not the adjacency one. It makes it also more comparable with the other models.

- ***Model D: FEMM:***

  $$\hat{A} = \sum_{\Delta t=0}^{+\infty} P\left(\Delta t\right) A^{\Delta t+1} \quad with \quad P\left(\Delta t\right) = \frac{e^{-\beta \Delta t}}{\sum_{\Delta t}^{+\infty} e^{-\beta \Delta t}}$$

  which can be re-written:

  $$\hat{A} = \left(1 - e^{-\beta}\right) A \left(I - e^{-\beta} A\right)^{-1}$$

  We then computed the average estimate for each of the conditions for each design. Only the FEMM (model D) had one free parameter in its equation. To remove this free parameter and make the model more comparable to the others, we used a previously estimated value of $\beta$=0.06 reported in the literature (*Lynn et al., 2020*). To confirm that this estimation corresponded to our data, we computed the correlation between the subjects' data and the predictions for $\beta$ ranging from $10^{-15}$ to $10^{15}$. We smoothed this correlation vector to avoid local variations and found a plateau of high correlation for $\beta = [10^{-4}; 10^{-1}]$ with a maximum for $\beta = 0.049$ (correlation 81%). Similarly, we computed the correlation between the FEMM and the hitting time estimation as a function of $\beta$. Here again, following the same procedure, we found a plateau of high correlation from $\beta = [10^{-4}; 10^{-1}]$ with a maximum for $\beta = 0.053$ (correlation = 99.3%). The two models can then be considered quasi-equivalent with the $\beta$ parameter considered in this paper (0.06).

- ***Model E: Hitting time:*** For this model, we approximated its value by creating 50,000 item-long streams corresponding to each graph and computing the average number of elements between each pair of stimuli. We took the inverse of this value to make it more directly comparable with the other models.

### Neural models

- ***Model F: CA1 similarity:*** We used the neural network and the procedure explained in *Schapiro et al., 2017*, originally published by *Norman and O'Reilly, 2003*. We did not change any parameter from this original study because our goal was to see how predictable this model was for our paradigms. We trained it 25 times on each of our graph structures (for each paradigm,

25 batches for 8 groups with syllables and 8 groups with tones: 25*8*2=400 replications). We then presented after each training each node as input in isolation and recorded the pattern of activity in the CA1 layer. To estimate the similarity in nodes' encoding, we computed the correlation between the pattern of activity in CA1 for pairs of elements. Finally, we then made predictions on our task by comparing the similarity between two nodes linked by our four types of transitions.

- ***Model G: Hebbian Learning with decay:*** This model aim to implement the FEMM computation with an adaptation of the Hebbian approach proposed for associative learning. To achieve that, we declared a layer of neurons with at least one neuron per node of the graph (it can contain more for generalization to bigger networks). The neurons started firing with an exponential decay corresponding to the FEMM decay for each sound in the sequence. Thus, if another sound was presented before the previous neuron stopped firing, several neurons encoding for different nodes co- fired simultaneously. It biased the estimation of TP between two elements. This co-firing behavior can be computed using Hebbian learning rule to update the weights between the neurons. This weight Matrix is then an estimation of the Free Energy Minimization Model that will converge as the length of the input stream increases. To estimate this model, we followed the same procedure as for the Hitting Time. We created 50 000 item-long streams corresponding to each graph and used those streams as inputs of the neural network. We updated the weight matrix at each step using Hebbian rule as described before. The weight matrix after the 50 000 items was used as an estimation of the model.

## Model comparison

To compare models and data, we considered all experimental paradigms together. To make it comparable with the two-forced-choice data, we normalized each design prediction by the model's value for *familiar within community* transitions. We then pooled all data from all paradigms and estimated the correlation between the data and the models' predictions using 5000 bootstrap re-sampling occurrences. The p-values were estimated by counting the percentage of bootstrap occurrences correlating more with one model compared to another. All the bootstrap occurrences and their correlation with each pair of models are presented in *Figure 4B*. Each dot represents one bootstrap occurrence. The distribution of these dots below and above the diagonal indicates the comparison between two models. The scatterplot's shape shows the correlation, independence, or anti-correlation between two models. This main analysis of data and model comparison have also been performed for each subgroup of data (first/second session; tones/syllables) and are presented in *Figure 4—figure supplement 1*. To try better differentiate communicability with the other models, we recomputed the same correlation analysis but restricted to conditions where communicability makes qualitatively different predictions (*new within* vs *familiar between* transitions in the sparse and high sparse designs). By doing so, we reduced most of the correlation between models and only tested for specific contradictory predictions. We again find that hitting time, FEMM, and Hebbian models are equivalent and better than the other models (see *Figure 4—figure supplement 2*).

## Conclusion

The results shown in this study reveal (1) community representation in the auditory domain; (2) the persistence of a biased, subjective TPs' representation after learning; and most importantly (3) pruning and completion effects allowing to build a parsimonious representation of the underlying network structure. TPs are thus not exactly encoded by the participants but biased in a way that can be predicted by the free energy minimization computation. Importantly, the same model might explain human sensitivity to local and high-level regularities without the need for specific models for each task.

More research is needed to characterize how and where such computations take place in the human brain and how this bias varies across individuals and with development. However, Hebbian rules in the cortex and/or hippocampus might be plausible candidates for a biological implementation of this analytical model. Finally, finding appropriate metrics to cluster graphs is a current research topic in applied mathematics (*Newman, 2006*). Thus, we believe that understanding the cognitive processes at stake when humans are exposed to such structured networks might provide insight to cognitively and biologically plausible computations.

## Acknowledgements

This research has received funding from the European Research Council (ERC) under the European Union's Horizon 2020 research and innovation program (grant agreement No. 695710 to GDL). We thank Stanislas Dehaene and Mathias Sablé-Meyer for discussions and remarks during the design and the interpretation of the experiment. We also thank the Foundation *Les Treilles* for supporting this work (LB).

## Additional information

### Funding

| Funder | Grant reference number | Author |
| --- | --- | --- |
| Horizon 2020 - Research and Innovation Framework Programme | 695710 | Ghislaine Dehaene-Lambertz |

The funders had no role in study design, data collection and interpretation, or the decision to submit the work for publication.

### Author contributions

Lucas Benjamin, Conceptualization, Data curation, Formal analysis, Visualization, Methodology, Writing - original draft, Writing – review and editing; Ana Fló, Conceptualization, Writing – review and editing; Fosca Al Roumi, Methodology, Writing – review and editing; Ghislaine Dehaene-Lambertz, Conceptualization, Supervision, Funding acquisition, Methodology, Writing – review and editing

### Author ORCIDs

Lucas Benjamin (iD) http://orcid.org/0000-0002-9578-6039
Ana Fló (iD) http://orcid.org/0000-0002-3260-0559
Fosca Al Roumi (iD) http://orcid.org/0000-0001-9590-080X
Ghislaine Dehaene-Lambertz (iD) http://orcid.org/0000-0003-2221-9081

### Ethics

All participants gave their informed consents for participation and publication and this research was approved by the Ethical research committee of Paris-Saclay University under the reference CER-Paris-Saclay-2019-063.

### Decision letter and Author response

Decision letter https://doi.org/10.7554/eLife.86430.sa1
Author response https://doi.org/10.7554/eLife.86430.sa2

## Additional files

### Supplementary files
• MDAR checklist

### Data availability

All Data and analysis are publicly available at https://osf.io/e8u7f/.

The following dataset was generated:

| Author(s) | Year | Dataset title | Dataset URL | Database and Identifier |
| --- | --- | --- | --- | --- |
| Fló A, Al Roumi F, Dehaene-Lambertz G, Benjamin L | 2022 | Data and Analysis for "Humans parsimoniously represent auditory sequences by pruning and completing the underlying network structure" | https://doi.org/10.17605/OSF.IO/E8U7F | Open Science Framework, 10.17605/OSF.IO/E8U7F |

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
