## [Editor Report]

This paper communicates important findings on the learning of local and higher order structures in auditory sequences and will be of interest to researchers studying statistical learning, learning of graph structures, and auditory learning. The strength of the evidence is convincing, including a compelling demonstration that humans do not encode objective transition probabilities and the implementation of a wide range of sequence learning models that have been proposed in the literature.

---

## [Decision Letter]

**Decision letter after peer review:**

[Editors’ note: the authors submitted for reconsideration following the decision after peer review. What follows is the decision letter after the first round of review.]

Thank you for submitting the paper "Humans parsimoniously represent auditory sequences by pruning and completing the underlying network structure" for consideration by *eLife*. Your article has been reviewed by 3 peer reviewers, one of whom is a member of our Board of Reviewing Editors, and the evaluation has been overseen by a Senior Editor. The following individual involved in the review of your submission has agreed to reveal their identity: Cameron Ellis (Reviewer #3).

Comments to the Authors:

We are sorry to say that, after consultation with the reviewers, we have decided that this work will not be considered further for publication by *eLife*. Despite excitement about the approach and many strengths of the work outlined by the reviewers below, we felt that the confound in the parsing results (as explained below) was problematic, and there were several other analysis and central framing concerns that led us to the conclusion that this paper would not cross the high bar for publication at *eLife*.

*Reviewer #1 (Recommendations for the authors):*

In this paper, participants are exposed to auditory sequences generated by graphs with community structure. Transitions between community nodes are sometimes left out during exposure, allowing tests of generalization to those unseen transitions. The authors find that participants are sensitive to the structure in general, as well as to the novel within-community transitions, indicating an understanding of the structure that goes beyond the directly-experienced information. They apply several theoretical and neural models to the data and find a range of matches to the empirical results. The best-fitting models are FEMM (Free-Energy Minimization Model) and Hitting Time, and the authors conclude that the mechanisms of those models may underly the patterns observed in humans.

The observation that participants choose unseen within-community transitions at a high rate is novel and a compelling demonstration that humans do not objectively encode transition probabilities in a stream of sounds. The many implemented and compared models are also a considerable strength of this work. However, I believe there is a confound in the pressing probability results, and I am also concerned that the behavioral data may be too noisy across participants to confidently test between some of the highly correlated models.

1) The pressing probability results (top row, Figure 3) are interpreted as evidence that the participants have learned the community structure and thus can parse the sequences at community boundaries. However, this effect can arise without there having been any learning: Within a community, stimuli are repeated many times before moving to the next community, which should result in stimulus adaptation. At the transition to a new community, stimuli are observed that have not been repeated as many times as recently, so simple adaptation can serve as a strong parsing cue. The paper that introduced this paradigm (Schapiro et al. 2013) included Hamiltonian paths (where every stimulus is visited exactly once) during the parsing task to avoid this confound, but this paper does not include that condition.

2) The authors acknowledge that the behavioral data are quite noisy across participants, requiring a very large sample to detect differences between conditions. Even with the large sample, many of the pairwise comparisons shown in the bottom row of Figure 3 are not significant. This raises concerns about whether a detailed test between correlated models is possible based on these data. My understanding is that the authors pooled data across all participants and designs and then did bootstrap resampling for statistical tests. I am concerned that this procedure is inflating the seeming reliability of small differences in the data, and sacrificing the ability to statistically generalize to the population. This particular dataset does not seem likely to allow reliable model comparison, at least between the top four or five models here, which are highly correlated.

3) I did not follow the reasoning for the argument that Hebbian learning must be cortical instead of hippocampal. There is a long history in the literature of considering Hebbian learning within the hippocampus.

4) I did not understand the design decision to always include a familiar within-community transition in the forced choice trials nor the analysis/display decision to set those options to 50% condition preference.

*Reviewer #2 (Recommendations for the authors):*

By testing statistical learning in auditory streams generated based on full and sparse community structures, the authors aimed to clarify what types of representations of structure arise. In order to disentangle different accounts regarding the nature of such representations, they contrast learners' preference for sound quadruplets containing within- versus between-community transitions that either were already presented during the stream or were never presented before. Predictions of 7 different models are outlined and correlated with the human forced-choice data. The main result is that learners show a bias in their representation of local transitions, making them sensitive to the high-order structure that characterizes the environment. This result is in line with previous findings in a different behavioral task and with the predictions of models that implement an accuracy-complexity trade-off.

Strengths:

Directly comparing community structures with different levels of sparseness provides a unique way of generating contrasting model predictions for models that generate highly comparable predictions in most learning situations. The results, especially those of the forced-choice task, are compelling.

The number of models that are directly compared is impressive and data visualizations do a very good job getting across the main conclusions for people without a modeling background.

Weaknesses:

The main result provides a conceptual replication of the finding by Lynn et al. (2020) in the visual domain. I do not think that the current work per definition has insufficient novelty, yet how the current findings relate to but also extend this previous work could be further clarified.

There is very little embedding of the current work within the existing literature. To exemplify, the authors write that "Many studies on sequence learning proposed different and not always compatible ad-hoc models to account for their results" (p. 4). This claim does not do justice to the modeling work that has been done in the domains of statistical learning and sequence learning (e.g., SNR, PARSER, TRACX) targeting specific conditions where models do differ in their predictions (e.g., phantom words).

Analyses in the manuscript itself focus only on the second forced-choice test, but it seems that the trajectory of how representations are formed over time (first vs. second forced-choice test) could also be modeled and could be highly informative. Data for the experiments with syllables and tones are collapsed but there seems to be a large difference in the learning trajectory for the two stimulus types (as reflected in figure 7), which currently remains unexplained.

– Authors like Friston might claim that not only learning of structure but also processes like decision-making and action selection can be understood as minimizing expected free energy. What could the finding that the FEMM model explains the current learning data very well say about the overlap between cognitive representations for very different tasks?

– Multiple statements seem in need of references. Some examples:

"… and their potential importance in language acquisition" (p. 3)

"Many studies on sequence learning proposed different and not always compatible ad-hoc models to account for their results." (p. 4)

"… the classical poverty of the stimulus argument" (p. 14)

– P(A|B) and later notations of adjacent and non-adjacent transitional regularities: Unless you specifically refer to backward transitional probabilities P(B|A) is the more intuitive form to denote the transitional probability of sequence AB, i.e., probability of B given that A has been encountered. Positional subscripts as used for Ngrams could also be used to clarify.

Methods

– Some methodological choices are not clearly motivated:

Why are only quadruplets used in the forced-choice task, and not also pairs?

Why is the judgement always with a familiar-within transition rather than contrasting the other conditions directly as well (e.g. familiar-between vs. novel-between or new-within vs. new-between)?

– What were the instructions participants received before performing the forced-choice task? Relatedly, how might the fact that there were two separate forced-choice tasks, with more active listening in between (now potentially with more awareness), have affected the results?

– "The press bar task during attentive listening showed high sensitivity, but it only allowed to test within vs between community transitions during learning and thus assess for pruning effect and clustering." (p. 15). Whereas I follow how these data are informative about clustering I am not clear on how they assess pruning.

Results

– Figure 3: the grey bar presents chance, but would it not make more sense to plot actual preference for familiar within-community?

Are results for the press task collapsed over the two blocks?

– For the analysis of key presses:

Is this test a significant difference in the difference scores (familiar-within vs. familiar-between)? Would a nonparametric cluster-based test not be a better option?

Parsing probability peaks after 1000 ms. Given that individual auditory stimuli, last 250 ms is it fair to say participants are sensitive to switching between communities (which might suggest they detect the between-community transition), or rather do they detect that they are in a new community after hearing several stimuli of the new community?

– P. 8 "In contrast, the New Within Community transitions were never rejected", unless I misunderstand the preference measure this should be "were rejected at chance", for half of the trials people prefer familiar within-transitions, the other half of these (no preference).

– p. 8 "No differences were found between the experiments using tones and syllables. Thus, data were merged in the following analyses." This should be supported by including basic results, preferably separately for the first and second forced-choice tests. (for example in the supplementary materials).

– One reasonable explanation for slower learning with syllables could be the prior knowledge individuals have about the structure of language (i.e., "linguistic entrenchment").

*Reviewer #3 (Recommendations for the authors):*

Benjamin and colleagues present a compellingly designed study to address a question currently interesting to the learning/memory community: how do we extract sophisticated structure from statistically regular input? I think the design is elegant, albeit similar to visual analogs. The sample size is high and the analyses are mostly sound. The biggest strength of the analyses is the breadth with which they surveyed different viable models and the surprisingly high model fits they achieved. I raise a few concerns that I believe the authors can likely address.

1. The nature of the forced choice model comparisons

The way that the authors compared their forced-choice data to the model predictions is central to their paper, but two fundamental ambiguities need to be resolved.

Firstly, the authors state that they pool the data across the experiment conditions. Does this mean concatenating the bootstrap average choices per choice lure and experiment condition, and then comparing those with the model? If so, state this explicitly.

Secondly, and more importantly, is the 'Familiar Within-Community' condition included in that correlation? Due to the nature of the forced choice the authors performed, this condition is always one minus the average for the lure condition. My understanding is that the authors choose to peg this value to 50% because it isn't clear what they should do otherwise. For instance, this could be the average of the three lure conditions, but those data points are not independent.

I think including this pegged value in the model comparisons is unfair because 1) this pegged value is arbitrary and not real data, and 2) this unfairly hurts Models A, B, C, and F, which predict that there will be a change in this condition under different levels of sparsity.

2. First vs. Second test epoch

I was surprised to read Figure 7 in the supplement. Until this point in the paper, I believed that the authors used both testing epochs for their analyses and that there were no differences between the tones and syllables conditions. In fact, the authors stated: "No differences were found between the experiments using tones and syllables. Thus, data were merged in the following analyses."

However, in the methods section and the supplement, they show that this was not the case. Figure 7 shows substantial and meaningful differences between the conditions in the first half but the results reported in the main text are only from the second half of the testing. It seems ad hoc to only include the last epoch of testing because 1) it seems they used both epochs for the press task, and 2) if the authors thought that 4 mins of passive listening weren't enough to max out learning, then they should have made the passive listening longer.

Compounding this, it is currently hard to interpret the difference between syllables vs. tones and first vs. second training epoch because no behavioral data is reported for these (akin to Figure 3).

I think learning effects are interesting and should be discussed. It is especially interesting that the syllables condition is so radically different. Syllables are the more typical stimulus used in auditory statistical learning tasks. In fact, tones typically lead to worse statistical learning (Schapiro, et al., 2014 is one example that comes to mind). Hence I think it is worth considering why there is such a drastic learning effect.

3. Conclusions about the brain from model comparisons

I believe that the authors overstate the brain-based conclusions that can be drawn from the model comparisons they perform. Model G uses associative learning to model the participant's choices. This is described as akin to cortical Hebbian learning, in contrast to the hippocampal learning in Model F. I think this juxtaposition is overstated given the nature of the modeling performed.

Modeling behavior can help elucidate the brain basis of phenomena, but I think this is difficult and ought to be done carefully. In this case, Hebbian learning is ubiquitous in the brain (and simple nervous systems). This lack of specificity means it cannot be easily used to discriminate the brain locus of community structure computation. Perhaps there is some reason to think that the exponential decay the authors include in Model G is specific to the cortex, but I am unaware of such a reason.

4. Compression

I am not sure about the relevance of compression to the work presented here. To start, the authors state in their abstract that "the brain does not rely on exact memories but compressed representations of the world". However, I am not sure how the results of this study contribute to our understanding of compression in memory. If the authors want to say that information is lost during memory encoding, I think that is an unnecessary point to make since it is uniformly agreed on. If the authors want to bring in concepts from information theory about compression, I think they need to be more precise since the type of compression they presumably mean is lossy compression (i.e., information is lost), but compression can be lossless (i.e., information is retained but reformatted). In fact, based on how the authors use this term elsewhere, I think chunking might be a better concept to refer to than compression.

In figure 5, the authors suggest that the participants might only learn and retain two chunks. I think this speculation is provocative, but I could not see any evidence in this paper to support this claim. This is peculiar because the authors could have treated this hypothesis as one of the models that they used to try to account for the participants' data. If they did that, I suspect this model would not excel because it assumes that participants treat the edge nodes the same as the non-edge nodes. From multiple studies published using this community structure design, edge nodes are treated differently. Hence, if I were to speculate on a 'compressed representation' that the participants might have, I would assume they have four chunks: a pair of edge nodes and non-edge nodes for each cluster.

Miscellaneous

5. Could the authors report how many of each transition the participants are exposed to during the passive learning and press task? It seems like there are a lot of possible transitions, so it is possible that even in the fully connected condition, their input is still sparse.

6. The authors have two exclusion criteria for the different tests (button presses and choices), which result in different sample sizes. However, if they failed the force choice attention check, I don't think they should be included in the key presses and vice versa. Instead, I believe the exclusions should be the union of these two exclusion criteria.

7. Why were twice as many participants excluded from the press task in the fully connected condition compared to the other conditions? Perhaps participants really did think there were fewer or more events? To address this, the authors could report the histograms for each condition before exclusions to see if those differences exist

8. Why is RT so delayed for the press task? I expect that participants should only take 500-750ms to respond, which suggests that perhaps participants are responding to something after the time locking is done here. What offset is the time locking to? Based on the data, I would guess it is time-locked to the offset of the edge node before the transition rather than the offset of the edge node after the transition.

9. The authors state that they did Bonferroni correction to test whether the likelihood of participants pressing a key after a transition has diverged from chance. Does this mean that every time point (millisecond?) is used as an independent sample in the Bonferroni correction? This doesn't seem plausible with the data shown in Figure 3 and is unnecessarily conservative.

10. What is the noise ceiling for the model fit? Lots of ways to find this but one would be to split the behavioral data in half and see how correlated they are. I ask because I suspect you are close to that ceiling, which is impressive

Regarding point 1:

I think that the authors should exclude the "Familiar Between Condition" values from the model comparison. I firmly believe that removing this condition will make the model comparisons fairer and may change the conclusions.

Regarding point 2:

The authors should more fully discuss the results of Figure 7 in the main text. They may also want to show the learning trajectory. Furthermore, I think they must report the forced choice data for the first testing epoch for syllables and tones separately, so it is clear what is driving the differences shown in Figure 7.

Regarding point 3:

I recommend drastically reducing the amount of commentary that compares the likely brain bases of the community structure computations. In particular, I think Model G can just be referred to as a biologically plausible alternative model to Model F.

Regarding point 4:

I recommend cutting all commentary about compression in the paper, but I am interested to hear justification otherwise.

If the authors wish to keep some of the compression content in, I think the authors could test my conjecture raised in point 4 by looking at the different nodes used in the 'New Between Community' lure trials. In particular, is there a difference when the 'New Between Community' is between an edge node and a non-edge node compared to when it is just between two non-edge nodes (vs. two edge nodes, i.e., 'Familiar Between Community')? I suspect there will be.

Miscellaneous recommendations:

1. Mention in the main text that these were online studies.

2. Ali Preston's work on transitivity is relevant to several points raised here, so she should be cited.

3. I think it would be helpful to explain the logic of Model F more fully. For instance, it would be beneficial to say what the CA1 layer is doing in the model and why you chose to look at it in particular (i.e., MSP.)

4. It would be helpful to expand on the differences between hitting time and communicability since they seem like they would produce predictions that are more similar to what you report.

5. Please report the ISI of the stimuli. They are 250ms long, but I didn't see any mention of an ISI, so I assume the items are back-to-back.

6. Good job releasing the data, although because it was uploaded as a zip, I couldn't download it without my client crashing. Consider breaking up the zip into smaller files.

[Editors’ note: further revisions were suggested prior to acceptance, as described below.]

Thank you for resubmitting your work entitled "Humans parsimoniously represent auditory sequences by pruning and completing the underlying network structure" for further consideration by *eLife*. Your revised article has been evaluated by Floris de Lange (Senior Editor) and a Reviewing Editor.

The manuscript has been improved but there are some remaining issues that need to be addressed, as outlined below:

1) Analysis of learning dynamics

Reviewer 1 points out that "they do not analyze learning dynamics. They report behavior from the press task aggregated across all exposure. Only the second testing epoch is used because participants "have not learned and thus have not stabilized behavior" in the first epoch. This presents a missed opportunity to study learning dynamics and evaluate how well the models fit those dynamics – an important piece of evidence for model fit."

2) Make explicit assumptions/limitations

Both reviewers ask you to make more explicit some assumptions/limitations in several instances.

3) Reviewer 2 suggest that "an additional experiment designed to disentangle those could further strengthen the impact of this work". Please treat this as a suggestion for future research, rather than a requirement for the revised version of this manuscript.

*Reviewer #1 (Recommendations for the authors):*

The authors did a good job of addressing my major concerns; however, in doing so they reinforced, in my opinion, a concern that reviewer 2 raised explicitly and I alluded to in my first round of review. Namely, are the authors capturing the dynamics of sequence learning, and, if not, what does that mean for their claims? Below I summarize my concern and then list a few minor issues that remain.

Throughout the paper and their response, the authors state that they are interested in understanding the dynamics of sequence learning and how hierarchical structure is acquired more generally. One of the most tantalizing outputs of this work is the possibility that they have a "general model of sequence learning". The task was explicitly designed to "have intermediate points to check the learning status" so they could study the dynamic aspects of learning, rather than examine it as a snapshot. Moreover, the FEMM is described as normative account of how learning should unfold in order to quickly generalize to an appropriate hierarchical structure.

However, I believe there are a few ways the authors fail to meet these goals.

First, they do not analyze learning dynamics. They report behavior from the press task aggregated across all exposure. Only the second testing epoch is used because participants "have not learned and thus have not stabilized behavior" in the first epoch. This presents a missed opportunity to study learning dynamics and evaluate how well the models fit those dynamics – an important piece of evidence for model fit.

As an aside, what evidence is there that the participants haven't learned? In both tones and syllable conditions, they show reliable evidence of preference (according to Figure S2) in the first pressing epoch, which indicates learning. In the press task there is evidence of learning in syllables and tones for the first epoch (according to the figure used to respond to Reviewer 2). As stated previously, it seems ad hoc not to include the first epoch in the analyses when the experiment was designed to include it.

Second, the authors argue that FEMM is an optimal way to overgeneralize to learn quickly. If so, it seems that FEMM should be excellent at approximating behavior when participants are initially learning. This is not the case for the syllable condition, where the authors argue more learning has to occur to overcome pre-existing biases.

Third, in the author's new and improved discussion about compression, they state that there is a stage after what is being tested here in which compressed representations form. At this stage, the FEMM would no longer apply, and so it is not a "general model of sequence learning".

I think the author's argument is that FEMM and their other high-performing models capture an initial stage of learning and that some other models (e.g. compression) will explain subsequent representations later in learning. If this is true, they ought to clarify this explicitly by stating their paper is a snapshot in the dynamics of learning. In doing so, they would need to revise their section "A general model of sequence learning" where they imply that the FEMM captures the entire learning process, instead of a portion. Moreover, they should be clear about what portion of the learning process they think FEMM accounts for: their normative claims about the value of overgeneralization suggest early stages of learning, but their exclusion of the first epoch suggests they are not testing early learning. Finally, I think the authors should still shorten their description of the compression section since its presence in the discussion implies that the data in the paper supported this hypothesis.

*Reviewer #2 (Recommendations for the authors):*

In general, the authors produced a stronger manuscript that addresses many of the concerns that were raised. The analysis (figure S2) of the syllable and tone tasks separately reassures me that, at least for Session 2, a similar behavioral pattern and model comparison result is observed for both tasks. Methodological choices are better motivated, and the adjusted visualization of the AFC results is a lot clearer. Finally, the split between theoretical models and possible neural implementation, in addition to the more careful brain-based conclusion, is also an improvement.

The novelty of the paper is now communicated more clearly, yet I do still share the concern that noisy behavioral data from a single type of learning measure (AFC) limit the possibility for strong conclusions based on model comparison. For many reviewer suggestions, the authors argue that their experiment was not designed to look at those comparisons. Especially for the highly correlated models that do a good just explaining the current AFC data (i.e. Hitting time, FEMM, Hebbian) I am left thinking that an additional experiment designed to disentangle those could further strengthen the impact of this work.

Currently, the AFC judgment is always one against a familiar-within transition "because all models postulate their correctness", but could such different contrasts not help to disentangle correlated models?

In their response to Reviewer 3, the authors mention that the difference between transitions between edge nodes and transitions between non-edge nodes is a prediction made by the FEMM. If this is a unique prediction of FEMM this seems worth testing.

Methodological remarks section (p. 20-21):

–Whereas I found the arguments counter the use of a Hamiltonian path convincing, I do think it would be good to explicitly acknowledge the potential confound outlined by reviewer 1 and the reason why you believe simple adaptation is not what is going on in the pressing probability results.

– "However, this second metric has a low sensitivity as only a few trials can be collected resulting in data variability that was compensated by a very large sample of participants (N=727)." Data variability seems strange phrasing, as true score variance is not a problem, maybe what is meant is high error variance?

Regarding prior knowledge of syllable sequences:

– The authors use both the phrasing "priors on syllable sequences" and "a prioris that syllables are ordered in words" (p. 31), I think the latter is a lot less precise and can better be avoided. The authors also write "This a priori does not exist with tones", but is that true given our exposure to music?

– The idea that participants have prior knowledge of syllable sequences affecting their ability to learn about new transitions between syllables is introduced without references, suggesting that it might be a new idea, whereas there is literature on this: see for example Siegelman et al. (2018)'s paper titled Linguistic entrenchment: Prior knowledge impacts statistical learning performance.

– The Participants section states that participants were recruited via social media but mentions nothing about their language background. The linguistic stimuli were French diphones. So far I assumed the experiment language was French, but maybe that was not the case?

---

## [Author Response]

[Editors’ note: the authors resubmitted a revised version of the paper for consideration. What follows is the authors’ response to the first round of review.]

Reviewer #1 (Recommendations for the authors):In this paper, participants are exposed to auditory sequences generated by graphs with community structure. Transitions between community nodes are sometimes left out during exposure, allowing tests of generalization to those unseen transitions. The authors find that participants are sensitive to the structure in general, as well as to the novel within-community transitions, indicating an understanding of the structure that goes beyond the directly-experienced information. They apply several theoretical and neural models to the data and find a range of matches to the empirical results. The best-fitting models are FEMM (Free-Energy Minimization Model) and Hitting Time, and the authors conclude that the mechanisms of those models may underly the patterns observed in humans.The observation that participants choose unseen within-community transitions at a high rate is novel and a compelling demonstration that humans do not objectively encode transition probabilities in a stream of sounds. The many implemented and compared models are also a considerable strength of this work. However, I believe there is a confound in the pressing probability results, and I am also concerned that the behavioral data may be too noisy across participants to confidently test between some of the highly correlated models.1) The pressing probability results (top row, Figure 3) are interpreted as evidence that the participants have learned the community structure and thus can parse the sequences at community boundaries. However, this effect can arise without there having been any learning: Within a community, stimuli are repeated many times before moving to the next community, which should result in stimulus adaptation. At the transition to a new community, stimuli are observed that have not been repeated as many times as recently, so simple adaptation can serve as a strong parsing cue. The paper that introduced this paradigm (Schapiro et al. 2013) included Hamiltonian paths (where every stimulus is visited exactly once) during the parsing task to avoid this confound, but this paper does not include that condition.

Indeed, over the 3 possible ways to traverse the graph (Random walk, Hamiltonian walk – each node seen once – and Eulerian walk – each edge crossed once -), we chose to present participants with random walks. Although Schapiro et al. (2013) showed that this choice had a limited effect on participants’ learning (this lack of effect was replicated by Lynn et al. 2020), we agree that this choice needs better justification and discussion:

1) The Hamiltonian path introduces an increased predictability of the sequence since stimuli already presented can no longer be presented. Learning a graph can then be fully explained with n-gram approaches and cannot disentangle between the different learning models proposed. Indeed, as a node can be visited only once at each passage in the community, the predictability of the next element increases as the journey goes on until a perfect predictability for the last element and for the transition to the other community (after having visited the 5th element of the community, the last one is perfectly predictable as well as the change of community) This would lead to periodic predictions of ngrams (cf. the model presented in Author response image 1) whereas a random walk keeps the prediction flat. Only a random walk, therefore, allows for rigorously disentangling a local TP (or ngrams) calculation from higher order calculations such as FEMM or Hitting Time.

**Author response image 1. sa2fig1:** 

2) The number of different Hamiltonian paths compatible with the structure decreases drastically with sparsity, and the very sparse community has only one possible Hamiltonian path given the input (each step is imposed because of the previous one) and thus the stream would become a trivial looped repetition of a twelve-tone pattern. An example is presented in Author response image 2. While the consequences of a Hamiltonian walk are less dramatic in the sparse model, the limited number of paths would still have induced a large number of pattern repetitions that are highly salient to humans (Barascud et al., Southwell et al., 2016 2018).

**Author response image 2. sa2fig2:** Example of the only Hamiltonian walk compatible with a high sparse community design. Given the first item, the full sequence becomes completely deterministic and repeats with loops of 12 elements.

3) Finally, Hamiltonian paths would lead to rhythmic switches between communities that could be used as a segmentation cue and bias the learning.

Concerning the random walk, it is true that on average the tones belonging to the same community are presented closer in time than those belonging to different communities but the length of the walk within one community can be short without repetition or without going through all the tones of the community, or longer with repetition of some tones at a random distance. There is therefore no consistency over time that could allow to capture a repetition pattern. Note also that the absolute frequency of each tone is equal within the stream avoiding long-term habituation effect and the local transition probability is flat avoiding the possibility to predict the next tone. Finally, the tones frequency was distributed between the two communities preventing a separation based on an auditory spectral partition.

“Stimulus (i.e. sensory) adaptation” or repetition suppression in electrophysiological recordings is observed in the case of the immediate repetition of the same stimulus, which is not done in our stream. However, some adaptation could occur because of stimulus prediction (Todorovic et al. 2012) that would then not be a confound but the actual phenomenon explaining the learning of the structure despite no possible prediction with TP only. These two types of adaptations notably differ in their timing, we are currently running MEG with this paradigm to better investigate these questions.

In any case, we would like to emphasize that the adaptation concern raised by the reviewer does not affect the two forced choice analysis (because elements were presented in isolation). It is this analysis which is the main analysis and novelty of our work. The pressing task was only used to keep participants attentive during the stream and to compare with the previous literature in which this task was used (Schapiro et al., 2013) The previous framing of the paper might have been ambiguous and we reframed this part.

We have now included all these elements in the Material and methods / Stimuli paragraph.

2) The authors acknowledge that the behavioral data are quite noisy across participants, requiring a very large sample to detect differences between conditions. Even with the large sample, many of the pairwise comparisons shown in the bottom row of Figure 3 are not significant. This raises concerns about whether a detailed test between correlated models is possible based on these data. My understanding is that the authors pooled data across all participants and designs and then did bootstrap resampling for statistical tests. I am concerned that this procedure is inflating the seeming reliability of small differences in the data, and sacrificing the ability to statistically generalize to the population. This particular dataset does not seem likely to allow reliable model comparison, at least between the top four or five models here, which are highly correlated.

The two forced choice data are indeed noisy. This is due to two main reasons : The difficulty of the task and the few trials presented to each subject to keep the experiment relatively short and compatible with online data collection. However, most pairwise comparisons with an expected high difference based on the FEMM Model are in fact significant. The nonsignificant comparisons mostly concern conditions for which the FEMM model predicts no or marginal differences.

Among all the models proposed in the literature that we wanted to compare, 4 are highly correlated: two theoretical models (Hitting Time, FEMM) and two neural implementations (Hebbian, CA1). We agree that it does not make much sense to compare theoretical and neural implementations, as they are not mutually exclusive and are not on the same Marr level (Computational theory vs Hardware implementation). Therefore, we split the theoretical models and neural implementation in two subplots in Figure 4.

We did not postulate a difference between Hitting Time and FEMM because these two models describe a similar property of the graph (distance between nodes) with two different approaches and are essentially similar (>99% correlations between the two models). We have rewritten the paragraph to make this point clearer.

Concerning the neural implementation models, Hebbian modeling and CA1, they are very similar for most conditions.

However, they differ qualitatively in a substantial way in their predictions regarding New Within and Familiar Between Transitions in the High sparse design for which the CA1 model is clearly at odds with our data. Furthermore, when we estimated the correlation with bootstrapping, both models ranked relatively high but the Hebbian approach was systematically higher (small effect size but high significance).

Regarding now Communicability vs. Hitting time and FEMM, we believe that the claim that Hitting Time and FEMM are both better models than communicability is fair considering our data. First the correlation is largely and significantly stronger for these two models compared to Communicability. But also, communicability is making predictions opposite to what was observed in the data. Indeed, according to Communicability, New Within should be significantly less accepted than Familiar Between transitions whereas we observed the contrary. Therefore, communicability is not a valid model of human behavior in this task.

To be more precise in our assumptions when testing the models, we redid the correlation analysis but limited it to conditions for which the four correlated models and communicability make qualitatively different predictions (New Within vs Familiar Between in Sparse and High Sparse communities). By doing so, we reduced most of the correlation between models and only tested for specific contradictory predictions. We again find that Hitting Time, FEMM and Hebbian models are equivalent and better than the other models (see Figure 4—figure supplement 1).

To conclude, although these models are indeed generally correlated, they have different predictions for some of the tested conditions allowing us to disentangle them, thanks notably to our large sample, the test of three different paradigms (Full, Sparse and High Sparse) and the careful bootstrap comparisons resampling subjects within a large pool. Because our goal was to compare between all the models proposed in the literature, it is the way they account for the relative performance pattern across conditions rather than the effect size per se which is important.

3) I did not follow the reasoning for the argument that Hebbian learning must be cortical instead of hippocampal. There is a long history in the literature of considering Hebbian learning within the hippocampus.

The discussion on this part was badly framed. We wanted to propose that (contrary to the hippocampus suggestion) Hebbian learning could be both cortical or hippocampal and not claim that it must be cortical. We re-framed the argument.

4) I did not understand the design decision to always include a familiar within-community transition in the forced choice trials nor the analysis/display decision to set those options to 50% condition preference.

Our goal was to measure the familiarity for each type of transitions. Because learning might remain implicit for most participants, we were afraid that familiarity ranking would not be sensitive enough. Therefore, we chose a two forced choice task and assess familiarity relative to a reference. The Familiar-within condition is ideal as a reference because first this condition has many possible transitions and second these transitions have correct local transitions and belong to the structure. Therefore all other conditions were contrasted with this one.

The Figure 3 plots were misleading because the first column did not represent data but chance level. It was done to be in congruence with the plots figuring the model. We rewrote this part and redid the plots to be clearer.

Reviewer #2 (Recommendations for the authors):By testing statistical learning in auditory streams generated based on full and sparse community structures, the authors aimed to clarify what types of representations of structure arise. In order to disentangle different accounts regarding the nature of such representations, they contrast learners' preference for sound quadruplets containing within- versus between-community transitions that either were already presented during the stream or were never presented before. Predictions of 7 different models are outlined and correlated with the human forced-choice data. The main result is that learners show a bias in their representation of local transitions, making them sensitive to the high-order structure that characterizes the environment. This result is in line with previous findings in a different behavioral task and with the predictions of models that implement an accuracy-complexity trade-off.Strengths:Directly comparing community structures with different levels of sparseness provides a unique way of generating contrasting model predictions for models that generate highly comparable predictions in most learning situations. The results, especially those of the forced-choice task, are compelling.The number of models that are directly compared is impressive and data visualizations do a very good job getting across the main conclusions for people without a modeling background.Weaknesses:The main result provides a conceptual replication of the finding by Lynn et al. (2020) in the visual domain. I do not think that the current work per definition has insufficient novelty, yet how the current findings relate to but also extend this previous work could be further clarified.

We thank the reviewer for the kind comments. The study by Lynn and colleagues nicely showed that community learning was compatible with a new model they proposed. Here, our goal was different, we wanted to disentangle between the many different models of sequence learning. It is for that reason that we developed this sparse design. We believe that the current study greatly extends previous findings as:

It shows the completion and generalization of missing dataIt disentangles between many different sequence learning models.It strengthens the bridge between two literatures that are often independently considered : sequence and network learning.It shows that the results are preserved with very fast sequence presentation, leaving less time for explicit decision making than in the original paradigms.It extends the previous results to the auditory domain (this replication of Lynn et al’s study concerned only the 1^st^ paradigm).

We made the goal and novelty of the study clearer in the text.

There is very little embedding of the current work within the existing literature. To exemplify, the authors write that "Many studies on sequence learning proposed different and not always compatible ad-hoc models to account for their results" (p. 4). This claim does not do justice to the modeling work that has been done in the domains of statistical learning and sequence learning (e.g., SNR, PARSER, TRACX) targeting specific conditions where models do differ in their predictions (e.g., phantom words).

We acknowledge that this sentence was not well formulated and rephrased it. We also added a full paragraph, to better explain the relationship between this work and previous modelling approach such as PARSER or TRACX. However, we want to point out that those models focus on how chunks are extracted from a stream, whereas here, we only study how familiarity between transitions is perceived. The two processes are not equivalent as we showed in Benjamin et al. (2022). Thus, the approach is different, which explains why we did not add those models to the comparisons. Furthermore, this type of models, based on chunk recognition, should by construction reject new within transitions generalizations, which were accepted by our participants. We added comments on these models in the main text.

Analyses in the manuscript itself focus only on the second forced-choice test, but it seems that the trajectory of how representations are formed over time (first vs. second forced-choice test) could also be modeled and could be highly informative. Data for the experiments with syllables and tones are collapsed but there seems to be a large difference in the learning trajectory for the two stimulus types (as reflected in figure 7), which currently remains unexplained.

Indeed, we focused only on the second forced choice test to build on the maximal learning performances, because the models predict familiarity only after learning. The models even assume learning from an infinite amount of data. However, a sufficiently long learning can be considered to converge to this asymptotic state.

Tones and syllables became indeed similar only at the second test session, and the learning trajectory seems different for both types of stimuli. We postulate that for syllables, because they are the building blocks of language, participants have stronger priors about syllable sequences than about tone sequences. For example, flat transition probabilities violate language structure and this was the main argument for proposing statistical computations as a crucial mechanism for infants to acquire language and notably to build their lexicon (Saffran et al., 1996). Subjects' performance on the first test after syllable streams was not consistent with any of the models (nor between paradigms), contrary to the other 3 datapoints which are also consistent with themselves. Thus the task is probably harder with syllables than tones due to the adjustment which needs to be done for syllables between the priors and the real structure of the stream.

We did not study the tone-syllable difference in this paper because we believe that the data and design are not well suited for this and that we would have to design a specific experiment to be able to model the resolution of a conflict between priors and structure, which would lead to learning.

We used syllables because the team is working on language acquisition, and aspects of this experiment were intended for use in neonates, whose performance is generally better when speech is used compared to non-speech stimuli.

– Authors like Friston might claim that not only learning of structure but also processes like decision-making and action selection can be understood as minimizing expected free energy. What could the finding that the FEMM model explains the current learning data very well say about the overlap between cognitive representations for very different tasks?

We thank the reviewer for this comment; however, we have no data to support or deny any claim in that direction. Thus, we believe that this point goes beyond the focus of our paper. Nonetheless, we think that sharing a common principle or mechanism does not imply that those representations of these domains necessarily overlap.

– Multiple statements seem in need of references. Some examples:"… and their potential importance in language acquisition" (p. 3)"Many studies on sequence learning proposed different and not always compatible ad-hoc models to account for their results." (p. 4)"… the classical poverty of the stimulus argument" (p. 14)

We added references.

– P(A|B) and later notations of adjacent and non-adjacent transitional regularities: Unless you specifically refer to backward transitional probabilities P(B|A) is the more intuitive form to denote the transitional probability of sequence AB, i.e., probability of B given that A has been encountered. Positional subscripts as used for Ngrams could also be used to clarify.

We have changed our notation and used positional subscript for more clarity and consistency in the text.

Methods– Some methodological choices are not clearly motivated:Why are only quadruplets used in the forced-choice task, and not also pairs?

This design choice was made for consistency with other studies of the PhD (L.B.), especially with the idea of comparing latencies and ERP in future electrophysiology work. We added a sentence to let the reader know.

Why is the judgement always with a familiar-within transition rather than contrasting the other conditions directly as well (e.g. familiar-between vs. novel-between or new-within vs. new-between)?

We used familiar within transition as a reference because all models postulate their correctness as it is both locally and globally congruent. Furthermore, there are many different usable transitions in this condition. We have added details in the text. See also answer 4 to R1.

– What were the instructions participants received before performing the forced-choice task? Relatedly, how might the fact that there were two separate forced-choice tasks, with more active listening in between (now potentially with more awareness), have affected the results?

They were only asked to choose which of the two quadruplets was most likely to belong to the previously invented sequence.

The effect of awareness and of more active learning is hard to test in such a task. For the tone group, it seems that it changed nothing. For the syllable group, it is hard to disentangle between a longer exposure or the fact that they change their priors about syllable sequences (see response above).

Remember that the data were acquired on-line and that it is impossible to verify the participants’ attention to the stimuli if long unanswered minutes elapse. Thus we preferred to have two shorter streams than a long one.

A passive (with no instruction and no task) MEG experiment is currently run to address those kinds of questions.

– "The press bar task during attentive listening showed high sensitivity, but it only allowed to test within vs between community transitions during learning and thus assess for pruning effect and clustering." (p. 15). Whereas I follow how these data are informative about clustering I am not clear on how they assess pruning.

It is true that they only assess for a difference between familiar within and familiar between but do not inform on the relative familiarity of each (familiar between < familiar within), which is what we referred as pruning in the text. We changed this sentence to avoid any confusion.

Results– Figure 3: the grey bar presents chance, but would it not make more sense to plot actual preference for familiar within-community?

The grey bar represented chance and all the other bars represented familiarity of the subjects with the tested condition compared to the reference (familiar within transitions). There was no trial in which familiar within transitions were contrasted with themselves (see response to R1).

We changed the figure because it was confusing.

Are results for the press task collapsed over the two blocks?

Yes, we collapsed both blocks (We have added a sentence in the method to mention it). For completeness we report in Author response image 3 the results for the two sessions and the two groups (tones and syllables) separately:

**Author response image 3. sa2fig3:** 

– For the analysis of key presses:Is this test a significant difference in the difference scores (familiar-within vs. familiar-between)? Would a nonparametric cluster-based test not be a better option?

We compared the average key press probability after familiar-between transitions (corresponding to 4 transitions) with each combination of 4 familiar-within transitions (each purple line visible on the graph). This made the comparison exhaustive.

Parsing probability peaks after 1000 ms. Given that individual auditory stimuli, last 250 ms is it fair to say participants are sensitive to switching between communities (which might suggest they detect the between-community transition), or rather do they detect that they are in a new community after hearing several stimuli of the new community?

Indeed it is difficult to tell with behavior only on such a fast design. However, a previous slow design (Schapiro et al., 2013) showed that participants were sensitive to the transition and did not need multiple items to press the key and signaled their detection of a change of community. The 2-forced-choice task also provided very limited context and yet showed a difference between within and between transitions. We are currently conducting a MEG experiment to better understand the time course of the effect (we do find a significant effect ~120ms after the community change, arguing for sensitivity to the transition more than for evidence accumulation).

– P. 8 "In contrast, the New Within Community transitions were never rejected", unless I misunderstand the preference measure this should be "were rejected at chance", for half of the trials people prefer familiar within-transitions, the other half of these (no preference).

Indeed, we corrected the sentence.

– p. 8 "No differences were found between the experiments using tones and syllables. Thus, data were merged in the following analyses." This should be supported by including basic results, preferably separately for the first and second forced-choice tests. (for example in the supplementary materials).

All the results from the 2-forced-choice are now reported in the supplementary results. We modified the sentence of the main text presenting the statistical analyses “We report the results at the end of the learning (second block). Because no difference was found between the groups using tones and syllables (unpaired t-test for each condition, all ps>0.2), the data of the tone and syllable groups were merged in the following analyses”.

– One reasonable explanation for slower learning with syllables could be the prior knowledge individuals have about the structure of language (i.e., "linguistic entrenchment").

Indeed, we believe that it is what happened (see above) and we included this hypothesis in the text.

Reviewer #3 (Recommendations for the authors):Benjamin and colleagues present a compellingly designed study to address a question currently interesting to the learning/memory community: how do we extract sophisticated structure from statistically regular input? I think the design is elegant, albeit similar to visual analogs. The sample size is high and the analyses are mostly sound. The biggest strength of the analyses is the breadth with which they surveyed different viable models and the surprisingly high model fits they achieved. I raise a few concerns that I believe the authors can likely address.1. The nature of the forced choice model comparisonsThe way that the authors compared their forced-choice data to the model predictions is central to their paper, but two fundamental ambiguities need to be resolved.Firstly, the authors state that they pool the data across the experiment conditions. Does this mean concatenating the bootstrap average choices per choice lure and experiment condition, and then comparing those with the model? If so, state this explicitly.

Yes, it is what we did, with one bootstrap sampling per paradigm. Because different participants have been tested with the three paradigms, we computed the average performance of one bootstrap resampling per paradigm and concatenate those to compare with the predictions of the different models.

Secondly, and more importantly, is the 'Familiar Within-Community' condition included in that correlation? Due to the nature of the forced choice the authors performed, this condition is always one minus the average for the lure condition. My understanding is that the authors choose to peg this value to 50% because it isn't clear what they should do otherwise. For instance, this could be the average of the three lure conditions, but those data points are not independent.

This part was unclear in the original paper. Our goal was to estimate the familiarity of each condition and we used a 2forced-choice task between conditions and a reference. We chose to use a forced-choice test because we thought it would be more sensitive than a familiarity ranking. We used as reference the Familiar Within Community transitions because all models postulate their correctness as they are both locally and globally congruent. Furthermore, there are many different usable transitions in this condition. This implies that all forced-choice tests consisted of one sequence with all Familiar Within Community transitions and one sequence containing a middle transition belonging to another type. While we did not have a test comparing two different Familiar Within Transition, this comparison can be assumed to be 50% by design. The other conditions can range from 0% (participants always preferred the reference) to 100% (participants always preferred the tested condition), with 50% implying no preference.

One minus the average of the other conditions would give the actual number of times the participant preferred the reference in the task but not an estimated familiarity with this condition. We changed the figure to better account for what we did.

I think including this pegged value in the model comparisons is unfair because 1) this pegged value is arbitrary and not real data, and 2) this unfairly hurts Models A, B, C, and F, which predict that there will be a change in this condition under different levels of sparsity.

Although we did not test the familiar-within community vs familiar within community condition, we can fairly assume that there are no differences between sequences of the same condition. We need this milestone in the correlations, because it is not only the relative familiarity between the three tested conditions which is important but also their familiarity relative to the familiar within-community (reference), in other words it is important to take into account that none of our conditions was preferred compared to the reference in our behavioral task. We effectively did not measure the true value of this condition by not measuring familiar within-community transitions versus other familiar within-community transitions, which should correspond to 50% more/less noise. We did include this condition in order to keep the experiment short while collecting as much data as possible to obtain more accurate measures of the familiarity of the other crucial conditions.

To illustrate the necessity of adding this milestone value in the models, we present two different fake models (see Author response image 4) that would not be differentiated if this 50% baseline value is ignored. In our dataset, it would, for example, artificially increase the correlation of the Non adjacent transition probability model with our data whereas the predictions of that model are quite different from the pattern observed in our data.

**Author response image 4. sa2fig4:** Two fake models are presented here that only differ in their predictions of the Familiarity score for the within community transitions (dark purple bar) compared to other conditions, that is, whether or not, its score is higher than the other conditions. If we consider only the three conditions on the right without their relationship to the purple bar, both models would be considered similar and equally correlated to our data but only Fake Model 1 is consistent with our observations (All conditions < at chance level). Fake Model 2 does not represent our data because all tested conditions are preferred to within-community transitions, thus would have be chosen in our forced-choice task. Therefore, the threshold value of Familiar within community transitions = chance level = 50% is essential to disentangle the models.

Regarding the second point, adding this value does not unfairly hurt models that predict a change in familiarity estimates of familiar within-community transitions as the models in each paradigm have been normalized based on this value to avoid it. Indeed, because each participant participates in only one of the paradigms (to avoid contamination of the learned structure to the next paradigm), there is no way to compare overall changes in familiarity between experiments. We therefore only sought to compare the estimate of familiarity within an experimental paradigm. For this reason, all models were also normalized relative to their familiar within community value before being concatenated for correlation (ModelValuesForCorrelation = ModelValues/FamiliarWithinCommunityValue). After normalization, all models estimates for the familiar within-community transitions was 1 for each of the three paradigms, thus not unfairly penalizing any model in the comparison with the data. To not include this value in the correlation would be to ignore the crucial information that the familiarity ranking of other transitions was actually lower than chance (<50%) in each paradigm.

2. First vs. Second test epochI was surprised to read Figure 7 in the supplement. Until this point in the paper, I believed that the authors used both testing epochs for their analyses and that there were no differences between the tones and syllables conditions. In fact, the authors stated: "No differences were found between the experiments using tones and syllables. Thus, data were merged in the following analyses."

We consider the measures at the end of learning. The results of the first session for the syllable group are uninterpretable while the other 3 data points (first and second session in the tone group and second session of the syllable group) are congruent. Participants had probably stronger priors about syllable sequences than about tone sequences. For example, flat transition probabilities violate language structure and this was indeed the main argument for proposing statistical computations as a crucial mechanism for infants to acquire language and notably to build their lexicon (Saffran et al., 1996). Thus the task is probably harder with syllables than tones due to the adjustment which needs to be done for syllables between the priors and the real structure of the stream.

We have corrected the sentence to avoid misunderstanding.

However, in the methods section and the supplement, they show that this was not the case. Figure 7 shows substantial and meaningful differences between the conditions in the first half but the results reported in the main text are only from the second half of the testing. It seems ad hoc to only include the last epoch of testing because 1) it seems they used both epochs for the press task, and 2) if the authors thought that 4 mins of passive listening weren't enough to max out learning, then they should have made the passive listening longer.

As this type of paradigm has never been done in the auditory domain, we were unaware of the time needed for successful learning neither of the difference between syllables and tones learning curve. We would like to point out that our models only predict the final learning for infinite sequences. It seems reasonable that human performances converges toward this with increased learning.

Furthermore, because in an on-line experiment during which it is difficult to ensure that participants are really doing the task, we wanted to avoid too long stream and have intermediate points to check the learning status.

Compounding this, it is currently hard to interpret the difference between syllables vs. tones and first vs. second training epoch because no behavioral data is reported for these (akin to Figure 3).

We added the 2-forced choice data for each of the group in the supplemental figure. Note that the results are very similar for 3 of the 4 blocks (1st session Tones, second session Tones and Syllables). Only the 1st session with syllable differs and does not present any reliable pattern with any of the possible models. Moreover, the pattern of results is not even consistent between the three paradigms, nor with any of the proposed models, making it most probable that a majority of participants have not learned and thus have not a stabilized behavior.

I think learning effects are interesting and should be discussed. It is especially interesting that the syllables condition is so radically different. Syllables are the more typical stimulus used in auditory statistical learning tasks. In fact, tones typically lead to worse statistical learning (Schapiro, et al., 2014 is one example that comes to mind). Hence I think it is worth considering why there is such a drastic learning effect.

Indeed the question of why the syllables and tones are different is an interesting point. However, since the experiment was not designed to disentangle different hypotheses on this aspect, we can only speculate on the possible reasons. As mentioned above, transition between syllables are constrained in speech and it is very likely that adults have internalized the transition matrix corresponding to their native language and first tried to apply this model to new data rather than rebuilding an entire model. Moreover in the daily life, sequences of syllables correspond to a succession of words, biasing participants to search for stable syllable transitions which were not present in the stream. Although we had no formal discussion with participants, some pilots/participants also reported that they were looking for familiar words hidden in a nonsense noisy sequence. These priors might explain the harder task in the case of syllables and that participants need some time before changing strategy and learn the real structure of the sequences.

We believe this hypothesis to be too speculative to make a strong point in the paper. However, we briefly discuss this idea in the text.

3. Conclusions about the brain from model comparisonsI believe that the authors overstate the brain-based conclusions that can be drawn from the model comparisons they perform. Model G uses associative learning to model the participant's choices. This is described as akin to cortical Hebbian learning, in contrast to the hippocampal learning in Model F. I think this juxtaposition is overstated given the nature of the modeling performed.

Indeed the cortical localization of the hebbian learning was overstated in the text, we corrected it (see response to R1).

Modeling behavior can help elucidate the brain basis of phenomena, but I think this is difficult and ought to be done carefully. In this case, Hebbian learning is ubiquitous in the brain (and simple nervous systems). This lack of specificity means it cannot be easily used to discriminate the brain locus of community structure computation. Perhaps there is some reason to think that the exponential decay the authors include in Model G is specific to the cortex, but I am unaware of such a reason.

We agree with that point, see response to the 3rd remark of reviewer1

4. CompressionI am not sure about the relevance of compression to the work presented here. To start, the authors state in their abstract that "the brain does not rely on exact memories but compressed representations of the world". However, I am not sure how the results of this study contribute to our understanding of compression in memory. If the authors want to say that information is lost during memory encoding, I think that is an unnecessary point to make since it is uniformly agreed on. If the authors want to bring in concepts from information theory about compression, I think they need to be more precise since the type of compression they presumably mean is lossy compression (i.e., information is lost), but compression can be lossless (i.e., information is retained but reformatted). In fact, based on how the authors use this term elsewhere, I think chunking might be a better concept to refer to than compression.In figure 5, the authors suggest that the participants might only learn and retain two chunks. I think this speculation is provocative, but I could not see any evidence in this paper to support this claim. This is peculiar because the authors could have treated this hypothesis as one of the models that they used to try to account for the participants' data. If they did that, I suspect this model would not excel because it assumes that participants treat the edge nodes the same as the non-edge nodes. From multiple studies published using this community structure design, edge nodes are treated differently. Hence, if I were to speculate on a 'compressed representation' that the participants might have, I would assume they have four chunks: a pair of edge nodes and non-edge nodes for each cluster.

We agree with the reviewer that this part of the discussion was not relevant it its previous form. However, in the discussion we would like to propose the hypothesis that the biased computation might be the basis of a later condensed abstract representation. We believe that what we observed in this study is not an abstract graph representation yet, but that this bias could lead (after sleep?) to an abstract compressed mental model only relying on two groups.

We have removed most references to compression in the main text and abstract and we have only mentioned it in a small paragraph in which we explain that it is a hypothesis for interpreting the utility of such a bias for humans. We also completely changed the Figure 5 to better reflect this line of thought. We hope that this part of the discussion is better written and should not be interpreted as a result but as a speculative hypothesis linking this study to the literature on abstract mental representation.

About the difference between transitions between edge nodes and transitions between non-edge nodes, it is a prediction made by the FEMM. However, in this study we did not have enough data per subject to divide each condition into these two subtypes and properly investigate this question.

Miscellaneous5. Could the authors report how many of each transition the participants are exposed to during the passive learning and press task? It seems like there are a lot of possible transitions, so it is possible that even in the fully connected condition, their input is still sparse.

The training stream is composed of 960 items (959 transitions) and each press task stream is composed of 480 items (2 times 479 transitions).

In the full community there are 48 individual transitions with the same probability, which represents an average of 20 presentations of each transition during the passive listening and 10 per press task which makes it quite unlikely to leave a sparse input. We added those values in the text.

6. The authors have two exclusion criteria for the different tests (button presses and choices), which result in different sample sizes. However, if they failed the force choice attention check, I don't think they should be included in the key presses and vice versa. Instead, I believe the exclusions should be the union of these two exclusion criteria.

We made two independent criteria because the number of button press in the press task could reveal a lack of attention or an inability to learn the structure. It seemed unfair to remove from the two-forced choice subjects because they could fail in learning the structure. However, this rejection criterion is not crucial, and the results remain similar when all subjects are included (see figure 3).

In the forced-choice task, we included catch trials to catch subjects that would press as fast as they could to make the experiment shorter or just randomly press without listening and we excluded these subjects as it is classical.

7. Why were twice as many participants excluded from the press task in the fully connected condition compared to the other conditions? Perhaps participants really did think there were fewer or more events? To address this, the authors could report the histograms for each condition before exclusions to see if those differences exist

We apologize for the error but the numbers reported in the paper were wrong. The real numbers are : FC 28/250, SC 24/249, HSC 23/228, which is close for all paradigms. We corrected it.

8. Why is RT so delayed for the press task? I expect that participants should only take 500-750ms to respond, which suggests that perhaps participants are responding to something after the time locking is done here. What offset is the time locking to? Based on the data, I would guess it is time-locked to the offset of the edge node before the transition rather than the offset of the edge node after the transition.

First, we have checked, and haven’t found any error in the timing reported, the data are time-locked on the offset of the transition. The task is quite hard for participants and they hesitate a lot in responding. We observe that the increase in the pressing behavior begins between 550 and 825 ms after the transition but is maximal around 1-1.2s after the transition.

9. The authors state that they did Bonferroni correction to test whether the likelihood of participants pressing a key after a transition has diverged from chance. Does this mean that every time point (millisecond?) is used as an independent sample in the Bonferroni correction? This doesn't seem plausible with the data shown in Figure 3 and is unnecessarily conservative.

We indeed used Bonferroni correction for 2851 points (time window from -100 to 2750 ms). The difference between MSC and the other paradigms is significant enough to pass this conservative approach. The lack of difference between FC and SC is not significant even without any correction (all ps>0.1) correction. The correction method does not impact the results here.

10. What is the noise ceiling for the model fit? Lots of ways to find this but one would be to split the behavioral data in half and see how correlated they are. I ask because I suspect you are close to that ceiling, which is impressive

We have added an estimation of the noise ceiling fit with the same bootstrapping approach. For each bootstrap, we randomly selected n subjects with replacement twice and correlated the data of those two random samples. We found an average of 84% correlations as a noise ceiling for those data. The maximum fit with FEMM is around 81% while the average of the bootstrap estimation with FEMM is around 77%.

We are indeed close to the ceiling as the model precisely predicts our data and a very large sample has been collected. We added this remark in the text and the figure.

Regarding point 1:I think that the authors should exclude the "Familiar Between Condition" values from the model comparison. I firmly believe that removing this condition will make the model comparisons fairer and may change the conclusions.

We re-explained why we used this 50% value and made clearer that it did not penalize models because each model was normalized based on this value.

Regarding point 2:The authors should more fully discuss the results of Figure 7 in the main text. They may also want to show the learning trajectory. Furthermore, I think they must report the forced choice data for the first testing epoch for syllables and tones separately, so it is clear what is driving the differences shown in Figure 7.

As suggested, we reported all the data in the supplementary material and mentioned the difference in the main text. As for the trajectory, unfortunately we believe that the experiment design is not suited to study this difference, which is also not related to the goal of the study.

Regarding point 3:I recommend drastically reducing the amount of commentary that compares the likely brain bases of the community structure computations. In particular, I think Model G can just be referred to as a biologically plausible alternative model to Model F.

We removed a full paragraph arguing for pattern completion and pattern separation in the hippocampus being the origin of the computation and just mentioned it. We also rephrased all the sentences comparing the models to avoid discussing cortical vs hippocampus computation. Instead we insisted that biologically plausible models exist for these kind of computations.

Regarding point 4:I recommend cutting all commentary about compression in the paper, but I am interested to hear justification otherwise.If the authors wish to keep some of the compression content in, I think the authors could test my conjecture raised in point 4 by looking at the different nodes used in the 'New Between Community' lure trials. In particular, is there a difference when the 'New Between Community' is between an edge node and a non-edge node compared to when it is just between two non-edge nodes (vs. two edge nodes, i.e., 'Familiar Between Community')? I suspect there will be.

As explained above, we drastically reduced the discussion on this point.

Miscellaneous recommendations:1. Mention in the main text that these were online studies.

Done.

2. Ali Preston's work on transitivity is relevant to several points raised here, so she should be cited.

Done.

3. I think it would be helpful to explain the logic of Model F more fully. For instance, it would be beneficial to say what the CA1 layer is doing in the model and why you chose to look at it in particular (i.e., MSP.)

We increased the description in the text.

4. It would be helpful to expand on the differences between hitting time and communicability since they seem like they would produce predictions that are more similar to what you report.

The prediction of hitting time and communicability are largely similar except for the strength of the generalization effect which is greater for hitting time. It leads hitting time to be significantly better correlated with our data than communicability.

5. Please report the ISI of the stimuli. They are 250ms long, but I didn't see any mention of an ISI, so I assume the items are back-to-back.

Yes they are. We added it in the method.

6. Good job releasing the data, although because it was uploaded as a zip, I couldn't download it without my client crashing. Consider breaking up the zip into smaller files.

We checked the folder that should be downloadable now.

[Editors’ note: further revisions were suggested prior to acceptance, as described below.]

The manuscript has been improved but there are some remaining issues that need to be addressed, as outlined below:1) Analysis of learning dynamicsReviewer 1 points out that "they do not analyze learning dynamics. They report behavior from the press task aggregated across all exposure. Only the second testing epoch is used because participants "have not learned and thus have not stabilized behavior" in the first epoch. This presents a missed opportunity to study learning dynamics and evaluate how well the models fit those dynamics – an important piece of evidence for model fit."2) Make explicit assumptions/limitationsBoth reviewers ask you to make more explicit some assumptions/limitations in several instances.3) Reviewer 2 suggest that "an additional experiment designed to disentangle those could further strengthen the impact of this work". Please treat this as a suggestion for future research, rather than a requirement for the revised version of this manuscript.

To answer reviewers’ comments:

1) We provide a simple model of theoretical learning dynamic during this kind of task with and without prior knowledge on the statistics between elements and reported it in our response. However, the experiment was not designed to test for this dynamic (only two time points, no control measure of the priors), and such an analysis would suffer from these limitations and would not change our main message. We revised the text to explain the message better and insist on the fact that sequence learning in general is not limited to statistical computation and our study and model is therefore not a general model of sequence learning but is limited to a general model of statistical learning in sequences.

2) In general, we tried to better delimitate the assumptions and limitations of this study. We provide further comment on when the model is appropriate (statistical computations in sequences). We also discuss its limitations (not taking priors into account, no precise exploration of the learning dynamic, risk of habituation confound, noisy behavioral data).

3) Regarding the discrimination between Hitting Time and FEMM, we compare the predictions of both models on 1,000 networks randomly sampled from space of 12 nodes networks, and look at the most different predictions between the two models. We showed that they are very similar in most situations and that the networks with different predictions were not well suited for experiments like the current one. Nevertheless, they may provide some clues on which directions should be taken for further investigations. We tried to articulate the two models in the text to show the conceptual similarities and better explain why FEMM and Hitting Time are measuring the same property of the network.

Reviewer #1 (Recommendations for the authors):The authors did a good job of addressing my major concerns; however, in doing so they reinforced, in my opinion, a concern that reviewer 2 raised explicitly and I alluded to in my first round of review. Namely, are the authors capturing the dynamics of sequence learning, and, if not, what does that mean for their claims? Below I summarize my concern and then list a few minor issues that remain.Throughout the paper and their response, the authors state that they are interested in understanding the dynamics of sequence learning and how hierarchical structure is acquired more generally. One of the most tantalizing outputs of this work is the possibility that they have a "general model of sequence learning". The task was explicitly designed to "have intermediate points to check the learning status" so they could study the dynamic aspects of learning, rather than examine it as a snapshot. Moreover, the FEMM is described as normative account of how learning should unfold in order to quickly generalize to an appropriate hierarchical structure.However, I believe there are a few ways the authors fail to meet these goals.First, they do not analyze learning dynamics. They report behavior from the press task aggregated across all exposure. Only the second testing epoch is used because participants "have not learned and thus have not stabilized behavior" in the first epoch. This presents a missed opportunity to study learning dynamics and evaluate how well the models fit those dynamics – an important piece of evidence for model fit.As an aside, what evidence is there that the participants haven't learned? In both tones and syllable conditions, they show reliable evidence of preference (according to Figure S2) in the first pressing epoch, which indicates learning. In the press task there is evidence of learning in syllables and tones for the first epoch (according to the figure used to respond to Reviewer 2). As stated previously, it seems ad hoc not to include the first epoch in the analyses when the experiment was designed to include it.

We agree with the reviewer that the claim participants did not learn is unfair, given the results in the key-press task. However, the two tasks (i.e. during and after the stream) are different: During the stream, they react to a change whereas after the stream, isolated quadri-element sequences are proposed for which participants must judge which of the two is more congruent with the language they heard. This part might elicit more interference from prior knowledge on the correctness of isolated segment (which in the case of syllables are close to words). Indeed, several experiments have shown how the native language affects statistical learning in adults (Elazar et al., 2022; Onnis and Thiessen, 2013; Siegelman et al., 2018). We modeled the dynamics of learning with FEMM in the presence and absence of priors (see the figure below). Moreover, in the case of syllables, human adults know a priori that syllables form words with a fixed order of syllables. In our experiment, the borders are between communities and, within communities, the elements appear randomly. In other words, language follows a completely different structure than the networks used here, which may hinder the learning of this type of graph when it is formed of syllables. This second type of prior is more difficult to model.

In the absence of priors, the learning dynamic can be modeled by comparing the FEMM prediction computed of a sequence of n elements compared to the real structure. In Author response image 5, we plotted the learning dynamic of the full community structure. The learning curve takes much longer to converge when we add strong priors on the TP matrix between syllables that are not congruent with the structure (for representation purposes with choose a prior strength of approximately 1000 elements: learning rate α = sequence length / 1000).

**Author response image 5. sa2fig5:** 

Our results show that previous experience can alter the learning dynamic; however, the two learning curves seem to converge. In our experiment, we cannot properly study participants’ learning curves because we only have two datapoints, and we did not design the experiment to specifically test the effect of priors (we have not used syllables with different transition probabilities in participants’ native language). Thus, further research is needed to address this point. Note that we used syllables because we wanted to test preverbal infants on a similar stream (see our work on statistical learning in neonates (Benjamin et al., 2022; Flo et al., 2022)). Infants respond better to speech stimuli than to non-speech stimuli, probably due to attentional biases and/or neural network dedicated to speech processing. Furthermore, they have not accumulated enough evidence about their native language to have strong priors at the age we test.We are now more cautious in the text about what we can and cannot conclude from these results and have added a paragraph in the methodological remarks section to explicitly state that it is a limitation of the study. We have also enhanced the reference to figure S2 in the main text so that readers interested in Tone/Syllable differences can easily refer to it.

Second, the authors argue that FEMM is an optimal way to overgeneralize to learn quickly. If so, it seems that FEMM should be excellent at approximating behavior when participants are initially learning. This is not the case for the syllable condition, where the authors argue more learning has to occur to overcome pre-existing biases.

This sentence referred to the case in the absence of priors. Indeed, we re-ran the dynamic modeling without prior but comparing FEMM and TP. Zooming in on the early dynamic, we can see that FEMM converges faster than TP and describes the structure better for short sequences (it has already inferred the communities while the TP model needs to see each transition). However, FEMM converges to a lower correlation with the true transition matrix because it has learned biased transitions (as shown by the pruning and completion effect).

**Author response image 6. sa2fig6:** 

It is true that our text was not explicit about what it referred to. Given the small size of the modeling difference and the fact that we did not directly test it, we did not include this test to avoid overinterpretation of a small difference but modified the text to be more specific.

Third, in the author's new and improved discussion about compression, they state that there is a stage after what is being tested here in which compressed representations form. At this stage, the FEMM would no longer apply, and so it is not a "general model of sequence learning".I think the author's argument is that FEMM and their other high-performing models capture an initial stage of learning and that some other models (e.g. compression) will explain subsequent representations later in learning. If this is true, they ought to clarify this explicitly by stating their paper is a snapshot in the dynamics of learning. In doing so, they would need to revise their section "A general model of sequence learning" where they imply that the FEMM captures the entire learning process, instead of a portion. Moreover, they should be clear about what portion of the learning process they think FEMM accounts for: their normative claims about the value of overgeneralization suggest early stages of learning, but their exclusion of the first epoch suggests they are not testing early learning. Finally, I think the authors should still shorten their description of the compression section since its presence in the discussion implies that the data in the paper supported this hypothesis.

Indeed, we agree with the reviewer that FEMM is not a general model of sequence representations in the brain but only a general model of how to extract statistical information from a continuous sequence. It says nothing about what how subsequent processes might use this information. We have corrected the text to make clearer the distinction between structure learning and its flexible use. We have limited the implications of our experiment to the statistical learning part only and reduced the paragraph on the compression hypothesis.

Reviewer #2 (Recommendations for the authors):In general, the authors produced a stronger manuscript that addresses many of the concerns that were raised. The analysis (figure S2) of the syllable and tone tasks separately reassures me that, at least for Session 2, a similar behavioral pattern and model comparison result is observed for both tasks. Methodological choices are better motivated, and the adjusted visualization of the AFC results is a lot clearer. Finally, the split between theoretical models and possible neural implementation, in addition to the more careful brain-based conclusion, is also an improvement.The novelty of the paper is now communicated more clearly, yet I do still share the concern that noisy behavioral data from a single type of learning measure (AFC) limit the possibility for strong conclusions based on model comparison. For many reviewer suggestions, the authors argue that their experiment was not designed to look at those comparisons. Especially for the highly correlated models that do a good just explaining the current AFC data (i.e. Hitting time, FEMM, Hebbian) I am left thinking that an additional experiment designed to disentangle those could further strengthen the impact of this work.

First of all, it should be noted that we have used all these models because most have been proposed in the literature and we wanted to be as complete as possible and underline for some of them their similarities.

Regarding FEMM and Hebbian models, it is not possible to disentangle them because this Hebbian model was designed as an implementation of the FEMM model using the Hebb rule for neural binding. Indeed, both models share the same linear mixture of all TPs orders with the same exponential decay.

Regarding FEMM and Hitting Time, both models capture the same graph property (average distance between two nodes), thus are also difficult to disentangle. Nevertheless, we tried to find if the properties of certain networks might help to differentiate them. As there is no analytical solution for Hitting Time in the general case, we used simulation to look for such a graph. We simulated 1000 networks randomly sampled from all possible 12-nodes graphs with some reasonable constraints for a cognitive experiment such as: respected connectivity (every node can be reached from any other) and no nearly complete or nearly empty network. Then we computed the two metrics and the correlation between FEMM and HT predictions. The distribution of the correlations is plotted in Author response image 7.

**Author response image 7. sa2fig7:** 

We can see that both metrics are always highly correlated. The network with the lowest correlation had a 53% correlation (red square in the first plot below). TP, FEMM, and Hitting Time matrices are shown below. We can see that the difference between FEMM and Hitting Time is almost entirely driven by the value at position [11,9], due to a single possible transition from node 11 to node 9 with a transition probability of 1 (hitting time is thus equal to 1, which is much larger than for all other transitions). After removing this outlier transition, the correlation rises to 96%. This is the case (at different degrees) for all graphs we could find with a correlation below 80%, suggesting that the difference between the models is restricted to “aberrant” cases in which a transition with a transition probability of 1 is present and driven by it, and thus does not apply to the general case.Thus, we believe that FEMM, hitting time and Hebbian learning are similar models, capturing the same properties of transitions in a network. We included the three in the paper because they refer to different formalisms. FEMM is more accurate and uses a β parameter that could account for possible differences between ages and populations. Hitting Time is easier to understand if one thinks in terms of sequences rather than statistics and networks. Hebbian learning is just a proposed implementation of FEMM with already described Hebb’s rule (different levels in Marr’s classification). There may be some formalism for differentiating FEMM from Hitting Time, but networks do not seem well suited for this purpose, as they almost always give similar predictions in both models.

For better readability of the paper, we have interchanged Hitting Time and FEMM in the model description and figure 2. We now present FEMM computation first and later present Hitting time as an alternative view of the same property but from a sequential perspective. We hope that this new organization of the paper will help in understanding the conceptual similarities of the two models. We have also modified the text in several places to better describe the two models and their similarities.

Currently, the AFC judgment is always one against a familiar-within transition "because all models postulate their correctness", but could such different contrasts not help to disentangle correlated models?

Unfortunately, most of the other contrasts between conditions are equivalent in all models. The only difference in this design could come from New Within vs Familiar Between in the High Sparse Community design only, where Hitting Time and FEMM (with β = 0.06) have slightly different prediction. However, FEMM with a slightly different β could have similar prediction on the New Within vs Familiar Between condition.

In their response to Reviewer 3, the authors mention that the difference between transitions between edge nodes and transitions between non-edge nodes is a prediction made by the FEMM. If this is a unique prediction of FEMM this seems worth testing.

Here again, by design the three correlated models inherit of this difference because it comes from the property of average distance between nodes in the network which is the common metric that all models approximate.

Methodological remarks section (p. 20-21):–Whereas I found the arguments counter the use of a Hamiltonian path convincing, I do think it would be good to explicitly acknowledge the potential confound outlined by reviewer 1 and the reason why you believe simple adaptation is not what is going on in the pressing probability results.

We have added a paragraph to explicitly warn the reader about this: *“However, due to the design reasons explained before, Halmitonian walks are not usable and thus we could not formally control for potential habituation effect in our design. The key-press results of this study (but not the 2-forced choice results) are therefore potentially subject to confounding by habituation.”*

– "However, this second metric has a low sensitivity as only a few trials can be collected resulting in data variability that was compensated by a very large sample of participants (N=727)." Data variability seems strange phrasing, as true score variance is not a problem, maybe what is meant is high error variance?

We corrected in the text.

Regarding prior knowledge of syllable sequences:– The authors use both the phrasing "priors on syllable sequences" and "a prioris that syllables are ordered in words" (p. 31), I think the latter is a lot less precise and can better be avoided. The authors also write "This a priori does not exist with tones", but is that true given our exposure to music?

We corrected in the text and avoided the second formulation. For tones, habituation to music might also lead to priors, however the different tones we use are not belonging to any musical scale in particular, avoiding familiarity with musical grammar. We changed the text accordingly.

– The idea that participants have prior knowledge of syllable sequences affecting their ability to learn about new transitions between syllables is introduced without references, suggesting that it might be a new idea, whereas there is literature on this: see for example Siegelman et al. (2018)'s paper titled Linguistic entrenchment: Prior knowledge impacts statistical learning performance.

We added this reference and two others to support this claim (Elazar et al., 2022; Onnis and Thiessen, 2013; Siegelman et al., 2018).

– The Participants section states that participants were recruited via social media but mentions nothing about their language background. The linguistic stimuli were French diphones. So far I assumed the experiment language was French, but maybe that was not the case?

It was indeed French participants only. We added this information in the text.

References

Benjamin L, Fló A, Palu M, Naik S, Melloni L, Dehaene‐Lambertz G. 2022. Tracking transitional probabilities and segmenting auditory sequences are dissociable processes in adults and neonates. Developmental Science. doi:10.1111/desc.13300

Elazar A, Alhama RG, Bogaerts L, Siegelman N, Baus C, Frost R. 2022. When the “Tabula” is Anything but “Rasa:” What Determines Performance in the Auditory Statistical Learning Task? *Cogn Sci* 46:e13102. doi:10.1111/cogs.13102

Flo A, Benjamin L, Palu M, Dehaene-Lambertz G. 2022. Sleeping neonates track transitional probabilities in speech but only retain the first syllable of words. Sci Rep 12:4391. doi:10.1038/s41598-022-08411-w Onnis L, Thiessen E. 2013. Language experience changes subsequent learning. Cognition 126:268–284. doi:10.1016/j.cognition.2012.10.008

Siegelman N, Bogaerts L, Elazar A, Arciuli J, Frost R. 2018. Linguistic entrenchment: Prior knowledge impacts statistical learning performance. *Cognition* 177:198–213. doi:10.1016/j.cognition.2018.04.011

Whittington JCR, Muller TH, Mark S, Chen G, Barry C, Burgess N, Behrens TEJ. 2020. The Tolman-Eichenbaum Machine: Unifying Space and Relational Memory through Generalization in the Hippocampal Formation. Cell 183:1249-1263.e23. doi:10.1016/j.cell.2020.10.024